# From Content to Knowledge: Lightning Fast Long-Video Understanding with Neural Knowledge Representations

Yuchen Guan [1]   Xiao Li [2]   Zongyu Guo [2]   Xiaoyi Zhang [2]   Xiulian Peng [2]   Chun Yuan [1]   Yan Lu [2]

## Abstract

We propose a new paradigm for long video understanding by treating a long video as a Neural Knowledge Representation (NKR). NKR represents video contents neither as a stream of tokens nor pre-organized databases, but as an individual small portion of network weights attached to the VLM backbone. The NKR weights are optimized to encapsulate the video's semantic content via a novel Agentic Knowledge Distillation (AKD) process, where an agent automatically synthesizes dense descriptions and question-answer pairs to distill the video's knowledge into the NKR. While AKD serves as a comprehensive, one-time encoding phase, the resulting NKR transforms the video into a portable, reusable asset. At inference, the lightweight NKR is mounted onto a frozen Vision-Language Model (VLM), enabling direct, query-based understanding without reloading or re-encoding the original video. This approach decouples video length from inference cost, offering high amortized efficiency for multi-turn video understanding. Experiments on the LVBench benchmark show our method achieves performance comparable to state-of-the-art approaches while reducing end-to-end latency by over two orders of magnitude, opening new possibilities for interactive long-video understanding.

## 1. Introduction

Multi-modal Foundation models are increasingly deployed as agents that solve open-ended tasks by understanding and interacting with human users, e.g., as chatbots, personal assistants, or information analyzers (Wang & Lu, 2025; Li

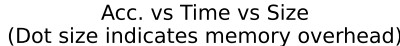

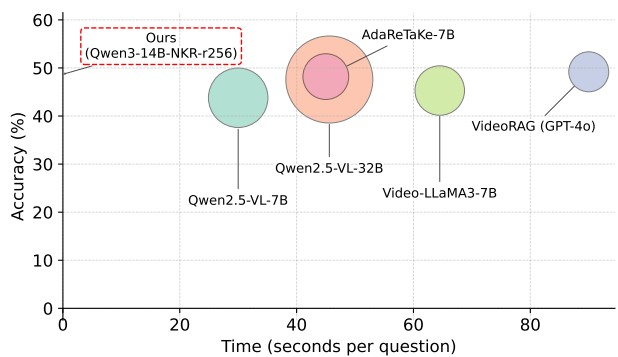

*Figure 1.* Time-memory-accuracy comparison on hour-level video understanding benchmark LVBench. The size of the circles indicates the additional memory overhead for video data during inference, measured in terms of the KV cache size for Qwen series, AdaReTaKe and GPT-4o, and external database size for VideoRAG. Compared to other methods, our proposed neural representation enables lightning fast response speed with nearly zero additional memory overhead during inference, while achieves similar accuracy.

et al., 2024). Among the capabilities such agent systems must have, long video understanding is both indispensable and uniquely challenging. On one hand, long videos distribute content sparsely from minutes to hours, making precise information extraction difficult. On the other hand, real-world scenarios often involve repeated querying and diverse downstream tasks on the same video, which necessitates low-latency responses and persistent, stable access to the video's underlying knowledge.

Currently, there are two dominant paradigms for video understanding using foundation models. The first directly tokenizes a video into a sequence of visual tokens fed to a Vision-Language Model (VLM) alongside the user query. Although recent VLMs perform well on short-video tasks using this strategy (Bai et al., 2025; Wang et al., 2025a), they still struggle with long video inputs (Wang et al., 2024a). Their key bottleneck arises from treating long videos as extremely long token sequences, leading to prohibitive computation and extra memory cost at runtime. In practice, limited context length and compute budgets often force aggressive frame sub-sampling or token pruning, thus discarding in-

---

[1]Tsinghua Shenzhen International Graduate School, Tsinghua University, Shenzhen, China [2]Microsoft Research Asia. Correspondence to: Xiao Li <xili11@microsoft.com>, Chun Yuan <yuanc@sz.tsinghua.edu.cn>.

*Proceedings of the 43rd International Conference on Machine Learning*, Seoul, South Korea. PMLR 306, 2026. Copyright 2026 by the author(s).

formation crucial for visual understanding. A second line of work adopts agentic pipelines that first organize a long video into a static database (*e.g.*, a RAG system), then iteratively plan, browse, retrieve, and reason over segments. While such systems have shown their better understanding performance on long videos (Zhang et al., 2025b; Chen et al., 2025), their explicit plan–act–observe loops introduce substantial latency that can be up to minutes per query, hindering the efficiency of multi-round interactions. In essence, existing methods fail to achieve the synergy of rapid responsiveness and long-term persistence, leaving the potential for truly efficient, multi-turn video interaction largely untapped.

In this paper, we jump out of existing paradigms and take a different stance of the video understanding problem. Our central observation is that efficient and reusable long-video understanding hinges on *what* knowledge is extracted and *how* it is represented. Here, *knowledge* refers to a higher level, comprehensive representation of the video content, which enables a model to recognize facts expressed in the video such as entities, events, attributes, relations, etc, and then apply logical and causal reasoning to infer new information that can answer user queries (Allen-Zhu & Li, 2023). We argue that enabling effective and efficient video understanding requires materializing a video's knowledge into an AI-native representation directly consumable by the model, instead of relying on dense sequence processing or external retrieval at runtime. Existing paradigms treat the video as a static, external "content container" that must be repeatedly opened and processed to extract query-aligned knowledge.

To this end, we propose a new paradigm for long-video understanding: the **Neural Knowledge Representation (NKR)**, which optimizes a video's entire semantic content into a compact, implicit neural representation. Our NKR materializes a video's knowledge into a small, swappable set of network parameters, analogous to how Neural Radiance Fields (NeRFs) (Mildenhall et al., 2020) compress a 3D scene into a network's weights. Instead of treating the video as an external "content container" that requires repeated access, our NKR internalizes the video's knowledge directly within the model's parameters. These parameters, which can be dynamically mounted onto a Vision-Language Model (VLM), serve as the AI-native representation of the video.

The NKR representation yields several fundamental advantages. First, the size of NKR is independent of video duration, avoiding progressive information attenuation as length grows. Second, inference now only requires loading the lightweight NKR adapter, bypassing the costly processing of video tokens or iterative database retrieval. Finally, the extra memory cost for video during inference is zero once the NKR is consumed into the VLM. Together, these properties enable highly efficient, lightning-fast responses for long-video understanding tasks.

Akin to other neural implicit representations, our paradigm performs offline optimization to compress the knowledge before online use. The key challenge is to construct an effective optimization objective that distills the rich knowledge of a video without human annotation. We design an *Agentic Knowledge Distillation* (AKD) process to address this challenge. Given an input video, AKD first conducts a holistic description extraction leveraging multiple foundation models and vision tools; it then synthesizes multi-level question–answer pairs covering entities, events, relations, timelines, as well as compositional reasoning questions requiring temporal localization and evidence aggregation. Finally, we optimize the NKR with a mixed-training strategy on both the descriptive data and the question answer data to distill the video into a constant-size implicit weight.

We evaluate our method on challenging long-video benchmarks including LVBench (Wang et al., 2024a) and LongVideoBench (Wu et al., 2024). Under comparable foundation model size, replacing direct video token input with our neural knowledge field yields a comparable overall performance and better long-video performance, while reducing response time by an order of magnitude. Our contributions can be summarized as:

- A new paradigm for long-video understanding that encodes a single video's knowledge into a **Neural Knowledge Representation (NKR)**.

- An **Agentic Knowledge Distillation** process that synthesizes supervision signals to distill video knowledge without human annotation.

- We demonstrated **lightning-fast responses with nearly zero memory overhead during inference** for long video understanding tasks.

## 2. Related Works

**Video Understanding.** Recent advances in foundation models (Achiam et al., 2023; Maaz et al., 2023; Zhang et al., 2023a; Wang et al., 2024d; Bai et al., 2025; Cheng et al., 2025a; 2026) have enabled video understanding by directly tokenizing video frames as long token sequences. While their capabilities have been adapted to video inputs with tens of frames, their performance degrades significantly on longer videos due to limited context window size (Wang et al., 2024a; Wu et al., 2024). Follow-up works have improved the context window by system/model co-design (Chen et al., 2024; Ding et al., 2024) as well as redundancy-aware token pooling/pruning methods (Wei & Chen, 2024; Wang et al., 2024b; 2025a). Yet, these methods still have the fundamental limitation of long-context processing efficiency, making it struggle to process hours-long videos. The rapid advancement of reasoning models (OpenAI, 2025) has also facilitated the development of agent

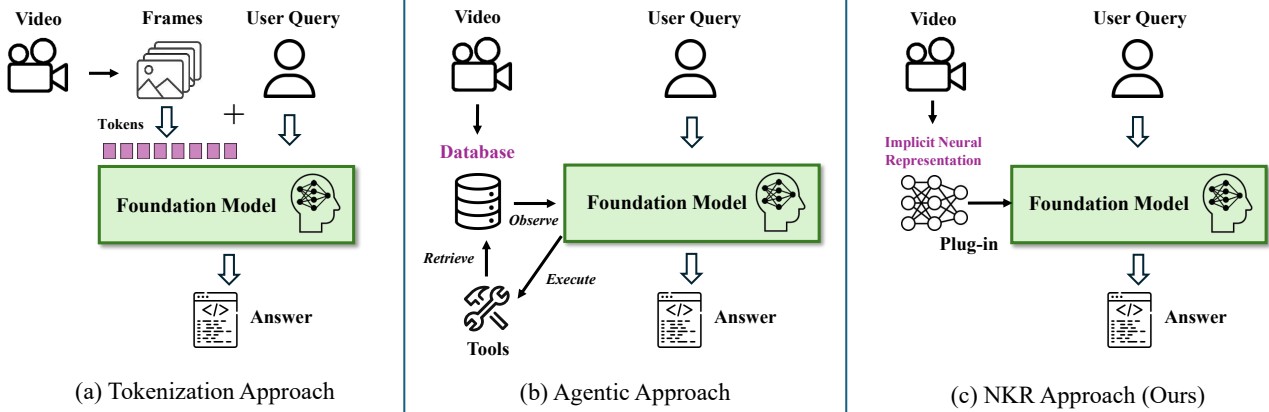

*Figure 2.* Different Paradigms for Long Video Understanding. (a) The commonly used approach for VLM treats video as token sequences; it is limited by the context window size for long video processing. (b) The agentic approach pre-computes the video into a static database that supports tool calling and retrieval; it requires iteratively query and execute tools, which severely limits its interactive response speed with users. (c) Our NKR approach optimizes a neural representation for high-level knowledge in the video; At run time, it is directly plugged into the foundation model, enabling lightning-fast responses to user query.

systems (Yao et al., 2023; Yang et al., 2023; Schick et al., 2023; Zhang et al., 2023b). A series of agent frameworks have focused on long-video understanding (Pang & Wang, 2025; Wang et al., 2025b; 2024c; Fan et al., 2024; Ren et al., 2025; Yan et al., 2025; Zhang et al., 2025b; Chen et al., 2025). Typically, these methods first decompose a long video into multiple short clips and pre-organize them as external databases. At runtime, the agent system iteratively thinks and conducts tool calls to retrieve relevant clips and generate answers. However, these approaches lead to extreme high latency, making them less suitable for interactive applications.

**Implicit Neural Representations.** Implicit neural representations (INRs), also known as neural fields, have emerged as a powerful paradigm for representing complex signals such as videos (Chen et al., 2021; He et al., 2026), 3D scenes (Wang et al., 2021; Mildenhall et al., 2020), light and reflection fields (Li et al., 2022; Wu et al., 2023; Xiong et al., 2024; Cheng et al., 2025b), etc. We refer to (Xie et al., 2022) for a comprehensive survey of INRs in the field of visual computing. Traditionally, INRs reconstruct signals by mapping continuous spatial coordinates to signal values using neural networks. The key advantage of INRs lies in their ability to compress high-dimensional signals with a compact set of parameters, enabling high-quality reconstruction and dynamic generation. Instead of using INR to reconstruct signal itself, we employ it to compactly represent the knowledge within long videos by mapping queries to relevant semantics, enabling efficient understanding and real-time query response with foundation models.

**Knowledge Distillation.** Knowledge distillation (KD) was originally proposed to transfer knowledge from a teacher model to a student model by matching softened output distributions (Hinton et al., 2015) or intermediate-

layer representations (Romero et al., 2015; Zagoruyko & Komodakis, 2016; Yim et al., 2017; Sun et al., 2019; Wang et al., 2020; Xu et al., 2020; Guan et al., 2024). In the foundation model era, KD has also been used to improve a student model's instruction following via supervised fine-tuning on data generated by a larger teacher (Wang et al., 2022; Taori et al., 2023; Xu et al., 2023), and to distill reasoning capability into smaller models by mimicking the teacher's intermediate reasoning or policy (Mukherjee et al., 2023; Shi et al., 2024; Guo et al., 2025; Wei et al., 2025; Xiong et al., 2025). Our approach also involves training a small part of model weights on data generated by a larger agent system but with a distinct goal: we aim to distill the internal knowledge of a media content itself (i.e., long videos) into a compact neural representation.

## 3. Method

In this section, we present our proposed new paradigm for long video understanding. We begin by briefly formulating the task and providing an overview of current approaches (Section 3.1). Next, we introduce our Neural Knowledge Representation (NKR) in (Section 3.2). Finally, we detail the Agentic Knowledge Distillation (AKD) process which optimizes a NKR module to effectively encode knowledge from a given long video input (Section 3.3).

### 3.1. Overview

Given a foundation model $\mathcal{M}$, a long video $V$ consisting of $T$ frames $\{f\} = \{f_1, f_2, \ldots, f_T\}$, and a user query $Q$ related to the video content, The goal of video understanding is to generate an accurate and contextually relevant response $A$ that addresses the query based on the information present in the video, i.e.,

$$A = \mathcal{M}(V, Q) \tag{1}$$

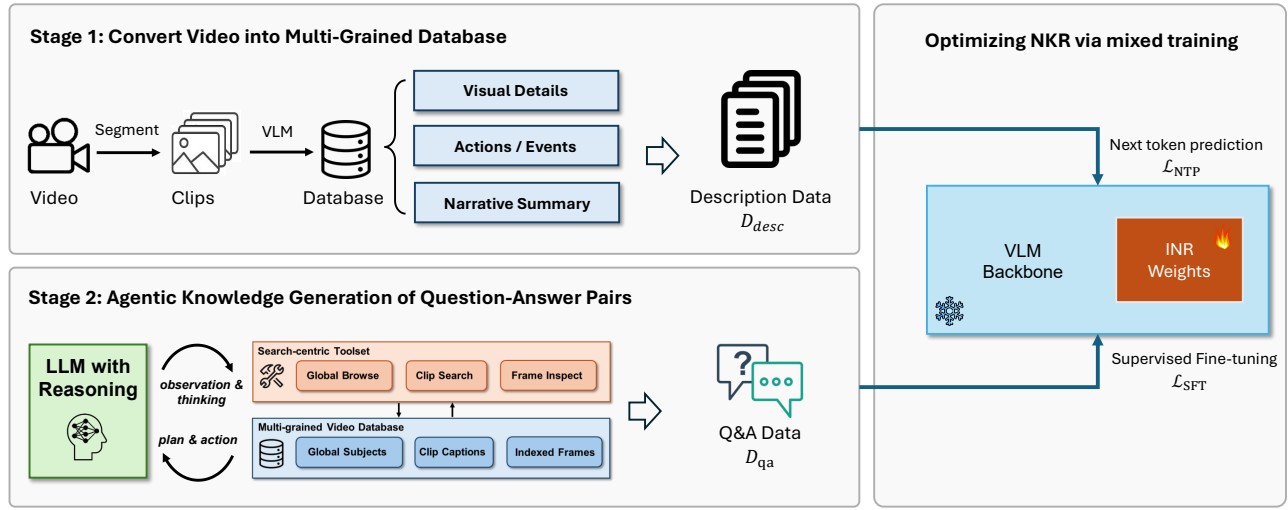

*Figure 3.* The agentic knowledge distillation (AKD) process for optimizing a neural knowledge representation (NKR). Left-up: given a video, we first segment it to multiple clips with different length and sample rates. Each clip is processed by a vision-language model (VLM) to extract multi-granularity textual descriptions and entity information, forming the dense description data $D_{desc}$. Left-bottom: we designed a ReAct-style agent to generate complex question-answer pairs $D_{qa}$, based on video content and dense descriptions. Right: the NKR is instantiated as a LoRA adapter for the given video, and optimized with a mixed-training strategy on both $D_{desc}$ and $D_{qa}$.

**Tokenized Approaches.** To handle video input, tokenized approaches (Bai et al., 2025; Lin et al., 2024; Hurst et al., 2024) work by first converting the video $V$ into a sequence of visual tokens $\{v\} = \{v_1, v_2, \ldots, v_N\}$ using a visual tokenizer $\mathcal{E}_v$, where $N$ is the number of visual tokens, and then feeding these tokens into the foundation model $\mathcal{M}$ along with the query $Q$:

$$A = \mathcal{M}(\{v\}, Q) = \mathcal{M}(\mathcal{E}_v(V), Q). \qquad (2)$$

While effective for image and short videos, tokenized approaches face significant challenges when dealing with long videos due to the limited long-context capability of foundation models. For example, GPT-4o (Hurst et al., 2024) and Qwen-2.5VL (Bai et al., 2025) has a maximum limit of 60 and 768 frames per query respectively, which is obviously insufficient for long videos.

**Agentic Approaches.** Agentic approaches (Ren et al., 2025; Zhang et al., 2025b) improve long video understanding by pre-organizing information from the video $V$ into a static database $\mathcal{K}(V)$, such as a RAG system. They then retrieve and process relevant information by iteratively calling a set of tools $\mathcal{T}$. The process can be described as a ReAct (Yao et al., 2023)-style loop:

1. **Plan:** Based on the query $Q$ and current context $S_i$, the reasoning model decides which tool $t \in \mathcal{T}$ to call and with which arguments.

2. **Act:** The tool $t$ is executed on the knowledge base $\mathcal{K}(V)$, producing a result $r_i$.

3. **Observe:** The result is added to the context: $S_{i+1} = S_i \cup \{r_i\}$.

This loop continues until a final answer $A$ can be synthesized from the accumulated context $S_{\text{final}}$. While agentic approaches substantially advance long video understanding, their gains require intricate planning and multi-round tool calls, sharply increasing time and computation costs. For instance, state-of-the-art approaches, Deep Video Discovery (Zhang et al., 2025b) repeatedly invokes the o3 reasoning model, usually requires 2–4 minutes to answer a single user query.

### 3.2. Neural Knowledge Representation

The limitations of both tokenized and agentic approaches stem from a shared, fundamental premise: they treat the video as an *external source of data* that must be processed or retrieved at inference time. Whether as a long sequence of tokens or a structured database, the video's content remains static and separated from the model's internal knowledge representation.

We argue that true efficient and deep understanding require a more integrated, AI-native representation of visual contents. Instead of (repeatedly) processing raw data during inference, we propose to *optimize* the entire knowledge of a given video into a compact, implicit neural representation that the foundation model can directly consume. Simply speaking, the input to the model is no longer a stream of visual tokens or an external database, but a set of learned parameters $\theta_V$ that **embody** the video's semantics.

**Neural Knowledge Representation.** Analogous to how Neural Radiance Fields (NeRFs) (Mildenhall et al., 2020) compress a 3D scene into the weights of a neural network for

novel view synthesis, we represent the video as an implicit neural knowledge representation (NKR) which condenses a long video into a compact, parameterized weights representation.

Formally, for a given video $V$, we aim to optimize a set of paramters $\theta_V$ that can be mounted into the given foundation model $\mathcal{M}$ at runtime. The resulting adapted model, denoted as $\mathcal{M}_{\theta_V}$, can then directly generate an answer $A$ for a query $Q$ without accessing the original video:

$$A = \mathcal{M}_{\theta_V}(Q). \tag{3}$$

Hence, the parameters $\theta_V$ serve as the materialized, AI-native representation of the knowledge contained within the input video $V$.

**Video Understanding with NKR.** The inference process with our NKR is highly efficient and consists of three simple steps: (1) *Load*: Instead of loading the video file, the system loads the lightweight neural weights $\theta_V$. (2) *Mount*: The weights are dynamically attached and merged to the frozen VLM backbone. (3) *Query*: The model directly processes the user's textual query $Q$ to generate the answer $A$. This approach offers two significant practical advantages. First, it enables **lightning-fast inference**, as it bypasses costly video token processing and iterative retrieval steps that are required by agentic methods. Second, the NKR has a **nearly zero memory overhead** once it has been merged with the VLM, effectively resolving the scalability issues of tokenized approaches.

**Parameter-Efficient Adaptation.** We implement our NKR using parameter-efficient fine-tuning (PEFT) techniques. Specifically, we employ Low-Rank Adapter (LoRA) (Hu et al., 2022) weights as our AI-native representation for the input video. The video-specific parameters $\theta_V$ are instantiated as a small set of LoRA weights attached to the attention layers of the VLM backbone. This design choice means that the core capabilities of the powerful VLM are preserved, while a small amount of parameters (i.e., the LoRA weights $\theta_V$) are optimized to specialize the model's behavior for the specific video input $V$. In our setup, changing video inputs simply involves swapping out the LoRA weights without modifying the foundation model $\mathcal{M}$ itself.

### 3.3. Agentic Knowledge Distillation

The core challenge in creating an effective NKR is to design an optimization strategy that can distill a long video into a compact set of parameters. An ideal NKR must satisfy two criteria: (1) it must comprehensively **memorize** the video's content, and (2) it must have the ability to **integrate** this memorized information to reason and respond to user queries when mounted on a VLM.

To achieve this, we introduce **Agentic Knowledge Distil-**

lation **(AKD)**, an optimization process that optimizes the NKR parameters $\theta_V$ to a given input video $V$. Figure 3 provides an overview of the AKD process. It consists of two key components: (1) an automated data synthesis agent that generates high-quality training data from the input video, and (2) a mixed-training strategy that combines self-supervised memorization with supervised fine-tuning to effectively distill knowledge into the NKR.

**Data Synthesis.** We design an automated agent pipeline to generate a comprehensive fitting dataset for a given video $V$. We generate two types of data for different purposes: a dense set of textual descriptions $\mathcal{D}_{\text{desc}}$ to facilitate memorization, and a diverse set of question-answer pairs $\mathcal{D}_{\text{qa}}$ to enable query-aware understanding.

First, we generate rich textual descriptions from the video to construct the dataset $\mathcal{D}_{\text{desc}}$. We adopt text-based strategy as most open-source VLMs suitable for training can only support text-based outputs. To construct the data, we first perform dense, multi-scale segmentation of the video $V$ into a large set of clips of different duration and sample rates. For each clip, we use a powerful VLM (GPT 4.1) with a carefully designed prompt to generate: (a) descriptions at low-level (visual details), mid-level (actions and events), and high-level (narrative summary) granularities; and (b) a summary of all salient entities (e.g., people, objects) present. For each generated text, we further augment it by rephrasing it multiple times using a faster LLM (GPT 4.1-mini). This diverse set of textual descriptions serves as a comprehensive proxy for the video's content. Optimization on this dataset forces the NKR to memorize and reconstruct the video's key information.

Second, to generate the question-answer dataset $\mathcal{D}_{\text{qa}}$, we repurpose a state-of-the-art video understanding agent *Deep Video Discovery (DVD)* (Zhang et al., 2025b) into a data generation agent. This agent iteratively performs in-depth analysis of the video to synthesize a wide variety of question-answer pairs at two levels:

- **Clip-level QAs**: To ensure broad coverage, we generate a large number of questions for each video clip. These are generated in a single pass by prompting a small reasoning model (o4-mini) to cover aspects like content summarization, entity retrieval, and event description.

- **Video-level QAs**: To ensure complexity and handle cross-clip questions, we generate a smaller set of challenging questions for the whole video. We conduct this by building a ReAct-style agent (Yao et al., 2023) using the most powerful reasoning model (o3). We build a RAG database over the video clips and iteratively uses tools for deep retrieval and visual content analysis, similar to Deep Video Discovery (Zhang et al., 2025b).

This process yields complex questions involving holistic understanding and long-term temporal correlation across the video.

Optimization on this dataset enables the NKR to effectively utilize its memorized knowledge to answer diverse queries. We provide detailed prompts, tool designs and implementation details for the data synthesis agent in the supplementary material.

**Mixed-training Strategy.** Inspired by recent findings on LLM training (Allen-Zhu & Li, 2023), we employ a mixed-fitting strategy that combines a self-supervised objective for memorization with a supervised objective for query-aware understanding. The full loss for optimizing the NKR is a combination of the next-token prediction loss $\mathcal{L}_{\text{NTP}}$ and a supervised fine-tuning loss $\mathcal{L}_{\text{SFT}}$:

$$\mathcal{L}_{\text{AKD}} = \mathcal{L}_{\text{NTP}} + \mathcal{L}_{\text{SFT}}, \quad (4)$$

The next-token prediction loss $\mathcal{L}_{\text{NTP}}$ is employed on the dense description data $D_{desc}$ to force the NKR to internalize the video's factual content, while the supervised fine-tuning loss $\mathcal{L}_{\text{SFT}}$ is employed on the QA data $D_{qa}$ to endow the NKR with the ability to transform its memorized, fragmented content into structured knowledge that directly addresses user queries.

**Implementation Details.** For a given video, we generate clips with log-linear durations from 5 seconds up to the video length to ensure multi-scale coverage. To augment the data, we rephrase each generated text 5 times using an auxiliary LLM. For QA generation, we generate 10 distinct QA pairs at the clip level, while our ReAct agent generates approximately 1000 complex questions at the video level. On average, a 1-hour video yields 30k–40k text description entries and 15k–20k QA pairs, with each ranging from 20 to 300 tokens. The data generation process for a 1-hour video takes 2–3 hours. The NKR is trained by alternately optimizing the next-token prediction and supervised fine-tuning on this synthesized data with a 2:8 ratio. Unless otherwise specified, we train a LoRA adapter with a rank of 256 on a Qwen3-14B model for 5 epochs. Training a 1-hour video takes approximately 2 hours on four 40GB A100 GPUs.

## 4. Experiments

### 4.1. Experiments Setup

**Datasets.** We evaluate the proposed NKR method on a long video understanding benchmark, LVBench (Wang et al., 2024a). The LVBench benchmark stands as one of the most comprehensive and challenging benchmarks for extreme long-form video understanding. It has a total of 103 videos with a total duration of 117 hours and 1549 manually annotated multiple-choice questions that require deep

*Table 1.* Examples of synthesized training data for a video clip.

Example Video (clip frames):

| Data Type | Example |
|---|---|
| *Data for Next-Token Prediction ($\mathcal{D}_{desc}$)* | |
| Low-level Desc. | From 00:00 to 00:15, In the opening moments, a female skater ... After dramatic lighting and overlays, the scene shifts to a blue Olympic stadium where the ice dance duo begins another performance ... The background is a vibrant Olympic venue in Vancouver 2010 ... Overlays present text such as 'COMPULSORY DANCE VANCOUVER 2010' ... Their choreography progresses through hold transitions, sweeping twizzles ... |
| Mid-level Desc. | From 00:00 to 00:30, A man and a woman, both wearing ice skates, are seen performing ... The woman wears ... The man ... then ... Their routine includes ... with visible text including ... |
| High-level Desc. | From 00:00 to 02:00, A male and female figure skating duo perform on ice at the Olympic Games. |
| Entity Summary | **Tessa Virtue**: female, athletic/slender build, fair/pale skin, Canadian ice dancer ... **Scott Moir**: male, athletic/medium build, fair/light skin, Canadian ice dancer ... **Audience**: large crowd in stadium, Olympic Games event spectators ... |
| *Data for Supervised Fine-Tuning ($\mathcal{D}_{qa}$)* | |
| Clip-level Q&A | **Q**: What exact phrase appears on the video's opening title card, setting the stage for the compilation? **A**: TESSA VIRTUE AND SCOTT MOIR EVERY PERFORMANCE AT THE OLYMPIC GAMES. |
| Video-level Q&A | **Q**: What is the primary purpose of the brief scene that opens the video at 00:00:00–00:00:10, showing Virtue in a pink dress and Moir in a grey shirt beneath an "Olympic Channel" overlay? **A**: It serves as an introductory montage announcing the compilation of all their Olympic performances. |

comprehension of the video's content, including temporal reasoning, event understanding and entity recognition.

**Baselines.** We compare the proposed NKR method with two existing dominant paradigms for long video understanding. For tokenization-based approach, we compare with the currently widely-used proprietary model GPT-4o and GPT 4.1 (Hurst et al., 2024), open-sourced VLM Qwen2.5-VL (Bai et al., 2025) and VideoLLaMA3 (Zhang et al., 2025a). We also compared with AdaReTaKe (Wang et al., 2025a) which greatly improves the input length of tokenization-based approach by KV-cache compression. For agentic approach, we compare with RAG-based agent VideoRAG (Ren et al., 2025), Tree-search based agent VCA (Yang et al., 2025), and ReAct-based agents Deep Video Discovery (Zhang et al., 2025b).

### 4.2. Main Results

Our main results on the LVBench benchmark is reported in Table 2, Overall, it demonstrates that the Neural Knowledge Representation (NKR) paradigm successfully redefines the trade-off between performance and efficiency in long video understanding. As shown our NKR approach achieves comparable overall accuracy (48.8%) while speed-up the infer-

*Table 2.* Accuracy, model size, input size, and inference speed comparison on LVBench benchmark. The input size refer to either max number of input frames or input video fps (whichever larger). Memory overhead is measured in GBytes; we measure the KV-cache size for tokenized approaches during inference and the size of the external database built upon the video for agentic approaches. our NKR weights are merged into the VLM at runtime, thus it has zero extra memory overhead. Inference speed is measured on a single NVIDIA A100 GPU with 80G memory for open-sourced models, while Qwen2.5-VL-32B uses 2×A100 due to its memory requirement. The question set of LVBench is categorized as follows: ER: Entity Recognition, EU: Event Understanding, KIR: Key Information Retrieval, TG: Temporal Grounding, Rea: Reasoning, Sum: Summarization. The inference time and memory overhead for VCA is not available because it does not have a public implementation.

| Methods | Input Statistics | | | Accuracy | | | | | | |
|---|---|---|---|---|---|---|---|---|---|---|
| | Input Size (fps or max frames) | Inference Time (s/query) | Memory Overhead (GB, at Runtime) | Overall | ER | EU | KIR | TG | Rea | Sum |
| *Tokenized approaches: Commercial VLMs* (Hurst et al., 2024; Team et al., 2023) | | | | | | | | | | |
| GPT-4o-20241120 | 60 frames | 43.0 | Unknown | 48.9 | 48.9 | 49.5 | 48.1 | 40.9 | 50.3 | 50.0 |
| GPT-4.1-20250414 | 50 frames | 31.7 | (Large) | 45.8 | 48.0 | 43.6 | 41.2 | 36.8 | 47.3 | 39.7 |
| *Tokenized approaches: Open-Source VLMs* (Bai et al., 2025; Wang et al., 2025a; Zhang et al., 2025a) | | | | | | | | | | |
| VideoLLaMA3-7B | 2fps (<768 frames) | 64.5 | 1.2 | 45.3 | 45.8 | 42.4 | 47.8 | 35.9 | 45.8 | 36.2 |
| Qwen2.5-VL-7B | | 30.0 | 2.5 | 43.8 | 43.0 | 41.9 | 49.8 | 40.5 | 43.8 | 32.8 |
| Qwen2.5-VL-32B | | 45.6* | 11.3 | 47.6 | 47.1 | 47.8 | 55.0 | 40.5 | 47.3 | 41.4 |
| AdaReTaKe-7B | 2fps (<2048 frames) | 45.0 | 0.9 | 51.2 | 51.1 | 47.6 | 62.2 | 43.2 | 50.2 | 27.6 |
| *Agentic approaches* (Ren et al., 2025; Zhang et al., 2025b; Yang et al., 2025) | | | | | | | | | | |
| VideoRAG | N/A | 90.0 | 0.5 | 49.2 | 47.4 | 49.3 | 57.1 | 36.5 | 43.9 | 39.7 |
| VCA | N/A | — | — | 41.3 | 43.7 | 40.7 | 37.8 | 38.0 | 46.2 | 27.3 |
| Deep Video Discovery-o3 | N/A | 180.0 | 0.3 | 74.2 | 73.4 | 73.3 | 80.4 | 72.3 | 70.7 | 74.1 |
| *Ours* | | | | | | | | | | |
| Qwen3-NKR-14B-r256 | N/A | 0.33 | 0 | 48.8 | 54.2 | 48.1 | 53.0 | 45.6 | 47.1 | 35.3 |

ence time by 100× than tokenization-based approaches with commercial GPT series (31.7s to 43.0s) open-sourced VideoLLaMA3 (64.5s) and Qwen2.5VL series (30.0s to 45.6s) as well as agentic methods such as VideoRAG (90.0s).

The efficiency advantage of NKR stems from a fundamental design choice: it decouples resource consumption from video duration compared to other approaches. Figure 1 visualizes the accuracy, speed, and memory cost of different methods. Tokenization-based methods handle longer videos by expanding visual tokens. The quadratic attention complexity leads to prohibitive increases in both computation time and memory usage (KV-cache). Agentic approaches also suffer from scalability issues, as longer videos enlarge the external databases, leading to slower retrieval and increased storage requirements. In contrast, our NKR paradigm eliminates the need for processing a perpetually growing size of video content during inference. Thus, NKR is the only method that (1) achieves both nearly instant inference speed and zero extra memory overhead regardless of video duration, and (2) maintains a competitive overall accuracy.

Table 3 further validates this scalability by analyzing video understanding performance w.r.t. duration. We partition the LVBench benchmark into two subsets of roughly equal size: videos shorter than one hour and those longer than one hour. While existing methods show significant performance

*Table 3.* Performance analysis w.r.t video durations on LVBench benchmark. Our NKR method demonstrates remarkable stability for both short and long videos.

| Method | < 1 hour | > 1 hour | Performance Drop |
|---|---|---|---|
| GPT4.1 | 48.7 | 42.0 | -6.7 |
| Qwen2.5VL-7B | 45.9 | 41.1 | -4.8 |
| Qwen2.5VL-32B | 51.6 | 45.3 | -6.3 |
| AdaReTaKe-7B | 51.4 | 48.4 | -3.0 |
| VideoRAG | 50.7 | 47.8 | -2.9 |
| Qwen3-NKR-14B-r256 | 48.9 | **48.6** | **-0.3** |

degradation on videos longer than one hour (e.g., a 6.7% drop for GPT-4.1 and a 2.9% drop for VideoRAG), our NKR's accuracy on long videos remains remarkably stable, decreasing by a mere 0.3%. This confirms NKR's superior scalability and its unique suitability for truly long-form video analysis.

Compared to agentic methods like VideoRAG, our NKR demonstrates a different but compelling capability performance. While VideoRAG has improved the accuracy in retrieval-heavy tasks (KIR: 57.1% vs. 53.0%), our NKR excels in reasoning-intensive questions (REA: 47.1% vs. 43.9%). This suggests that by internalizing knowledge, NKR fosters a more holistic understanding conducive to reasoning, rather than relying on multi-round, discrete retrieval. Meanwhile, our NKR achieved this comparable performance while being 300x faster and requiring no external database during inference.

*Table 4.* Ablation study on the impact of VLM backbone size and LoRA rank on NKR performance and speed. The main model used in other experiments is highlighted. Speed is measured on a single NVIDIA A100 GPU with 80G memory.

| VLM Backbone | LoRA Rank | Overall Acc. (%) | Speed (s/query) |
|---|---|---|---|
| Qwen3-8B | 256 | 39.5 | 0.26 |
| **Qwen3-14B** | 64 | 44.2 | 0.33 |
| | **256** | **48.8** | |

*Table 5.* Ablation study on data components generated via Agentic Knowledge Distillation. The experiment is conducted on Qwen3-NKR-14B-r256.

| Data Components | | | Accuracy (%) |
|---|---|---|---|
| Dense Desc. | Clip-level QA | Video-level QA | |
| ✓ | | | 14.3 |
| ✓ | ✓ | | 45.0 |
| | ✓ | ✓ | 38.8 |
| ✓ | ✓ | ✓ | 48.8 |

Finally, Deep Video Discovery (DVD) achieves the best accuracy on LVBench in our experiments. However, its design philosophy and target scenarios differ fundamentally from ours. DVD's performance stems from a powerful external reasoning model with iterative planning, which incurs even more inference latency (avg. 3 minute for a single question), making it more suitable for offline analysis rather than interactive applications. The goal of our work is not to pursue maximum accuracy at all costs. Instead, we aim to explore a novel paradigm that guarantees minimal and constant inference cost, while pushing the boundaries of understanding within this efficiency-first constraint. Consequently, our NKR framework should be seen as pioneering a different path: one that prioritizes fast response time and scalability, offering a compelling alternative for interactive applications where low latency is a critical requirement.

### 4.3. Discussions

**Ablation Studies.** To understand the key to NKR's performance, we conducted ablation studies on its core components: the NKR representation capacity and the Agentic Knowledge Distillation (AKD) process.

Table 4 reveals that the performance of NKR is more determined by the capacity of the VLM backbone rather than the size of the LoRA adapter. Upgrading the VLM from 8B to 14B yields a substantial 9.3% accuracy with only minor speed overhead (0.33s vs. 0.26s), whereas quadrupling the LoRA rank from 64 to 256 provides a more modest 4.6% improvement without increasing inference time. This result suggests that NKR acts as an efficient "knowledge adapter" whose effectiveness hinges on a powerful VLM to reason upon the distilled knowledge. The choice of a 14B model with r=256 thus offers an optimal balance of high

*Table 6.* Comparison of performance on subset of MME-RealWorld image understanding benchmark.

| Method | Perception | Reasoning | Overall |
|---|---|---|---|
| Qwen2.5-VL-7B | 39.9 | 24.6 | 32.1 |
| Qwen2.5-NKR-7B-r32 | 38.4 | 32.5 | 35.4 |

performance and minimal speed impact.

The multiple type of data generated from our AKD process is also indispensable. Table 5 shows that training on dense descriptions or QA pairs alone leads to significant performance drops. Without descriptions, the model only memorizes QA formats, failing to generalize. Without all QA pairs, it possesses facts but lacks the ability to integrate and reason over them. Without video-level QA pairs, the NKR failed to learn the complicated correlation of the video knowledge at a holistic level, leading to a noticeable performance drop as well. This demonstrates that AKD's strategy of combining holistic descriptions with structured reasoning examples is the key to successfully distilling complex video content into a functional neural representation.

**Image Understanding.** To further validate the effectiveness of our implicit representation, we evaluate NKR on image understanding tasks. We construct a diverse subset from MME-RealWorld (Zhang et al., 2024) with 38 images and 526 QA pairs (258 perception, 268 reasoning). The subset covering all original sub-tasks with at least two images for each question type. Each image's NKR is optimized using dense descriptions and 350 synthesized QA pairs via the same AKD procedure used for videos. As shown in Table 6, NKR improves overall accuracy by 3.3% over the Qwen2.5-VL-7B baseline. The advantage is particularly pronounced in reasoning tasks, where NKR achieves a 7.9% gain, highlighting its strength in complex cognitive understanding.

## 5. Conclusion

We have proposed representing a video as a Neural Knowledge Representation (NKR) for video understanding. Instead of treating videos as external data sources, our approach encodes the knowledge of a long video into a compact set of network weights. This is achieved through our proposed Agentic Knowledge Distillation (AKD) process, which automatically synthesizes a comprehensive dataset of textual descriptions and question-answer pairs from the video. By mounting the learned, video-specific NKR onto a frozen Vision-Language Model, we enable long video understanding with lightning-fast response and zero memory overhead at runtime while achieving comparable performance to current approaches. We believe our proposed NKR paradigm opens up new possibilities for how we interact with and process large-scale multimedia content with foundation models.

## Acknowledgements

This work is supported by the SSTIC Grant (KJZD20230 923115106012, KJZD20230923114916032, GJHZ20240 218113604008).

## Impact Statement

This paper presents work whose goal is to advance the field of Multimodal Understanding. There are many potential societal consequences of our work, none of which we feel must be specifically highlighted here.

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

# A. Limitations.

As same to any current LLMs and VLMs, the NKR cannot be guaranteed to output without any hallucinations. Building a NKR from video input currently still relies on upfront optimization for each video. A major limitation is the reliance on textual-only descriptions $D_{desc}$ for the AKD process. This design choice is due to the fact that current VLMs do not have the ability to *output* visual tokens as a supervision signal. Consequently, our NKR may not fully capture fine-grained visual details that are difficult to articulate in text. In our early experiments, we have attempted to add visual tokens only as *input* during the optimization process, (i.e., includes a video captioning task during training), which significantly increased the optimization time but yielded no significant improvements to the text-only counterpart. Overcoming this "textual bottleneck" and enabling direct distillation of visual knowledge is a significant challenge, not just for NKR but for the broader field of implicit knowledge representation. We believe that future advances in generative VLMs will unlock the potential for a truly multi-modal knowledge distillation, paving the way for even richer and more capable NKRs.

# B. Future works.

Future work includes extending this paradigm to other modalities, and investigating interactive scenarios with user feedback to further refine and adapt the knowledge representation.

# C. Implementation Details

### C.1. Dense Description

We prompted the OpenAI GPT-4.1 model to generate dense descriptions. The detailed prompts used in our dense description generation are illustrated in Figure S5 and Figure S6.

### C.2. Clip-level Q&A for Video NKR

To generated clip-level Q&A data, we prompted the OpenAI o4-mini model to generate question-answer pairs in a single pass, based on the dense description of each video clip as input. Figure S7 demonstrates the detailed prompts used in our clip-level Q&A generation.

### C.3. Q&A for Image NKR

The prompt for image Q&A generation is similar to the clip-level video Q&A generation. Figure S8 demonstrates the detailed prompts used in our image Q&A generation.

### C.4. Video-level Q&A for Video NKR

Our video-level Q&A generation agent follows a ReAct-style agentic pipeline, similar to Deep Video Discovery (DVD) (Zhang et al., 2025b). We roughly follow the overall pipeline design of DVD, with specific modifications on prompts as well as improvements on database constructions and toolsets design to better fit our Q&A generation task.

**Overview.** Specifically, given a long video $V$, our agent consists of a multi-grained video database $\mathcal{D}$ with timestamp indexing, a set of toolset $\mathcal{T}$, and a strong reasoning foundation model $M$ (in our case, OpenAI o3 (OpenAI, 2025)). The overall pipeline is illustrated in Algorithm 1. We categorize the generated video questions into the same six category defined in LVBench (Wang et al., 2024a). Figure S9 demonstrates the general instruction of our video-level Q&A generation agent; the specific instructions for each question categories are illustrated from Figure S10 to Figure S11, respectively.

**Multi-grained Video Database.** The multi-grained video database $\mathcal{D}$ consists of three components: (1) The raw video clips split from the long video $V$ with timestamps; (2) Dense descriptions generated in Section C.1 for each video clip; (3) An entity list extracted from the dense descriptions, which contains the main characters, objects, locations, and other key entities appearing in the video. We follow the same procedure as in (Zhang et al., 2025b) to extract and deduplicate the entity list. We associate each video clips (along with their corresponding dense descriptions) and the entity list with both semantic-based and timestamp-based indexing schemes. The semantic embedding for each clip is generated from the OpenAI's text-embedding-3-large model (ope), which allows the agent to retrieve relevant video clips and entities based on semantics. The timestamp-based indexing is maintained with an interval tree data structure (Halbert et al., 2025); it enables

---

**Algorithm 1** Agentic Video-level Q&A Generation.

---

**Input:** Video database $D$, LLM $M$, tool set $\mathcal{T}$, stop sign STOP, question category $C$, max steps $N$.
**Output:** a QA pair $\{Q, A\}$ for video $V$.
Initialize history $H_0 \leftarrow \{C\}$
**for** $i \leftarrow 1$ to $N$ **do**
   $R_i \leftarrow M.\text{REASON}(H_{i-1})$
   $T_i \leftarrow M.\text{CALL}(R_i, H_{i-1})$ where $T_i \in \mathcal{T} \cup \text{STOP}$
   **if** $T_i$ is STOP **then**
     **break**
   **end if**
   $O_i \leftarrow T_i(\mathcal{D})$
   $H_i \leftarrow H_{i-1} \cup \{(R_i, A_i, O_i)\}$
**end for**
**return** $\{Q, A\} = M.\text{FORMAT}(H_i)$

---

the agent to access video clips and entities directly based on their temporal occurrence in the video.

**Toolsets.** We provide the agent with four tools: *Global Browse*, *Clip Search*, *Temporal Search* and *Frame Inspect*. The *Global Browse*, *Clip Search*, and *Frame Inspect* tools are adapted from the original design of DVD (Zhang et al., 2025b) and we refer to their paper for more details. The *Temporal Search* tool is designed to help the agent directly retrieve video clips and entities based on timestamps, implemented by calling the intervaltree object with `tree.overlap(begin_t, end_t)`. Table S7 summarizes the action space of our agent.

*Table S7.* Action space overview of our QA generation agent. The first four actions are from our toolset and the final STOP action is designed as a stop criterion.

| Action | Parameter |
|---|---|
| GLOBAL BROWSE | video database $\mathcal{D}$ 
 user query $Q$ |
| CLIP SEARCH | video database $\mathcal{D}$ 
 agent synthesized query $\hat{Q}$ 
 return top-$k$ captions |
| FRAME INSPECT | raw video $\mathcal{V}$ 
 agent synthesized query $\hat{Q}$ 
 temporal range $[t_s, t_e]$ |
| TEMPORAL SEARCH | video database $\mathcal{D}$ 
 temporal range $[t_s, t_e]$ |
| STOP | Stop the process and return |

## C.5. Training Details

We use the Azure OpenAI service to access API of the GPT-series models for data generation and evaluation. For training, we utilize the LLAMA-Factory framework (Zheng et al., 2024) for training both Video NKR and Image NKR. The training hyperparameters for video and image NKR are summarized in Table S8.

*Table S8.* Hyperparameters for training the Neural Knowledge Representation (NKR).

| Hyperparameter | Video | Image |
|---|---|---|
| *Model Configuration* | | |
| VLM Backbone | Qwen3-14B | Qwen2.5-VL-7B |
| LoRA Rank ($r$) | 256 | 32 |
| LoRA Alpha ($\alpha$) | 512 | 64 |
| *Training Configuration* | | |
| Optimizer | AdamW | AdamW |
| Learning Rate | 2e-4 | 1e-4 |
| LR Scheduler | Cosine Annealing | Cosine Annealing |
| Warmup Ratio | 0.05 epoch | 0.1 epoch |
| Global Batch Size | 32 | 2 |
| Per-device Batch Size | 4 | 1 |
| Gradient Accumulation Steps | 2 | 1 |
| GPU Specification | NVIDIA A100-40G | NVIDIA A100-40G |
| Number of GPUs | 4 | 2 |
| Epochs | 5 | 3 |
| Weight Decay | 0.1 | 0.0 |
| Mixed Precision | bf16 | bf16 |

## D. Additional Experiments

### D.1. Results on LongVideoBench

To provide a more comprehensive comparison with existing methods on long video understanding, we further evaluate Video NKR on the LongVideoBench (Wu et al., 2024) benchmark. The LongVideoBench benchmark contains 6678 questions from 3763 videos, covering a wide range of video durations from a few seconds to an hour. In particular, we focus on the longest subset of the benchmark, which includes videos with durations ranging from 900 seconds to 3600 seconds (denoted as the LongVideoBench-Long). This subset has 564 questions from 188 videos. All evaluations are conducted only using video content (i.e., without using audio narrations or subtitles).

Table S9 summarizes the results of Video NKR and other baseline methods on the LongVideoBench-Long set. The results demonstrate similar findings to experiments on LVBench in the main paper: Video NKR achieves similar or even superior performance on long video understanding tasks while being significantly more efficient in terms of computation and memory usage.

### D.2. More Analysis on NKR

**Knowledge Validation.** To validate that our NKR framework indeed encodes content into a generalizable knowledge representation that LLMs can effectively leverage for complex understanding tasks, we conduct two additional analyses on image understanding task.

First, we perform a comparative experiment on image understanding to demonstrate that NKR serves as a more effective input format for LLMs than raw content. We establish a strong textual baseline where a complete, detailed description of an image is directly fed into an LLM. As our NKR is optimized from textual descriptions, this comparison is intentionally designed to be fair: by using the same source information (textual descriptions and text-based Q&A pairs), we isolate the impact of the representation format. Specifically, we design the textual baseline by concatenating all dense descriptions and the generated Q&A pairs of an image into a single long text as in-context examples. Then we prompt the LLM to answer the MME-RealWorld test questions based on this long text input. We refer to this baseline as the "text-based LLM". The text-based LLM baseline achieves an accuracy of 19.6% on the MME-RealWorld test set described in the main text. In contrast, our Image-NKR, optimized from the very same textual descriptions, reaches a significantly higher accuracy of 35.4%. Notably, the performance gain is most pronounced on the reasoning subset of questions (17.2% vs 32.5%). The results strongly indicate that an implicit neural representation is a more potent and suitable format for LLMs to perform

*Table S9.* Accuracy, model size, input size, and inference speed comparison on LongVideoBench-Long benchmark. The input size refer to either max number of input frames or input video fps (whichever larger). Memory overhead is measured in GBytes; we measure the KV-cache size for tokenized apporaches during inference. our NKR weights are merged into the VLM at runtime, thus it has zero extra memory overhead. Inference speed is measured on a single NVIDIA A100 GPU with 80G memory for open-sourced models, while Qwen2.5-VL-32B uses 2×A100 due to its memory requirement.

| Methods | Input Size (fps or max frames) | Time (s/query) | Memory Overhead (GB, at Runtime) | Accuracy |
|---|---|---|---|---|
| GPT-4o-20241120 | 60 frames | 43.0 | Unknown (Large) | 60.9 |
| Qwen2.5-VL-7B | 2fps | 30.0 | 2.5 | 45.2 |
| Qwen2.5-VL-32B | (<768 frames) | 45.6* | 11.3 | 48.4 |
| AdaReTaKe-7B | 2fps (<2048 frames) | 45.0 | 0.9 | 55.5 |
| Qwen3-NKR-14B-r256 (Ours) | N/A | 0.33 | 0 | 52.9 |

complex understanding tasks.

Second, we investigate whether NKR's success stems from mere memorization of answers to highly similar questions in the synthesized Q&A pairs used during optimization. To test this, we perform a k-nearest neighbor search, using the test questions as queries over the entire generated Q&A dataset ($\mathcal{D}_{qa}$). Figure S4 demonstrates that the retrieved results are semantically related but not identical to the test questions. This suggests that the performance gains arise from NKR's ability to generalize knowledge, i.e., to activate relevant internal knowledge to answer a given query, rather than simply recalling previously seen pairs.

**Ablation of Different Teachers.** To validate the robustness of the proposed NKR against variations in the QA synthesis process, we employ three models with varying capabilities (GPT-4o mini, GPT-4.1, and GPT-5.4) to generate 150 textual samples each. We then train our model on the image dataset mentioned in the main paper using these synthesized texts. As presented in Table S10, the results demonstrate that our approach maintains strong robustness across different QA synthesis models.

*Table S10.* Ablation study on teacher models generated via Agentic Knowledge Distillation. The experiment is conducted on Qwen3-NKR-8B-r32.

| Method | Perception | Reasoning | Overall |
|---|---|---|---|
| GPT-4o mini | 37.0 | 44.3 | 40.3 |
| GPT-4.1 | 31.9 | 46.4 | 38.4 |
| GPT-5.4 | 34.9 | 45.9 | 40.5 |

**Comparison with In-Context Baseline.** To validate the effectiveness of the NKR paradigm, we further compare its performance against an in-context baseline that is provided with the exact same textual data (150 textual samples), either injected into the model's context window or absorbed via NKR. As shown in Table S11, NKR substantially outperforms the in-context baseline across both models, with a particularly pronounced gap on the Reasoning subset (e.g., +15.3 points for Qwen2.5-7B and +37.1 points for Qwen3-8B). This suggests that NKR learns a more generalizable internal representation rather than merely recalling matched text.

*Table S11.* Performance comparison between NKR and the in-context baseline on Qwen2.5-7B and Qwen3-8B.

| Method | Perception | Reasoning | Overall |
|---|---|---|---|
| Qwen2.5-InContext-7B | 22.1 | 17.2 | 19.6 |
| Qwen2.5-NKR-7B-r32 | 38.4 | 32.5 | 35.4 |
| Qwen3-InContext-8B | 24.4 | 9.3 | 17.6 |
| Qwen3-NKR-8B-r32 | 31.9 | 46.4 | 38.4 |

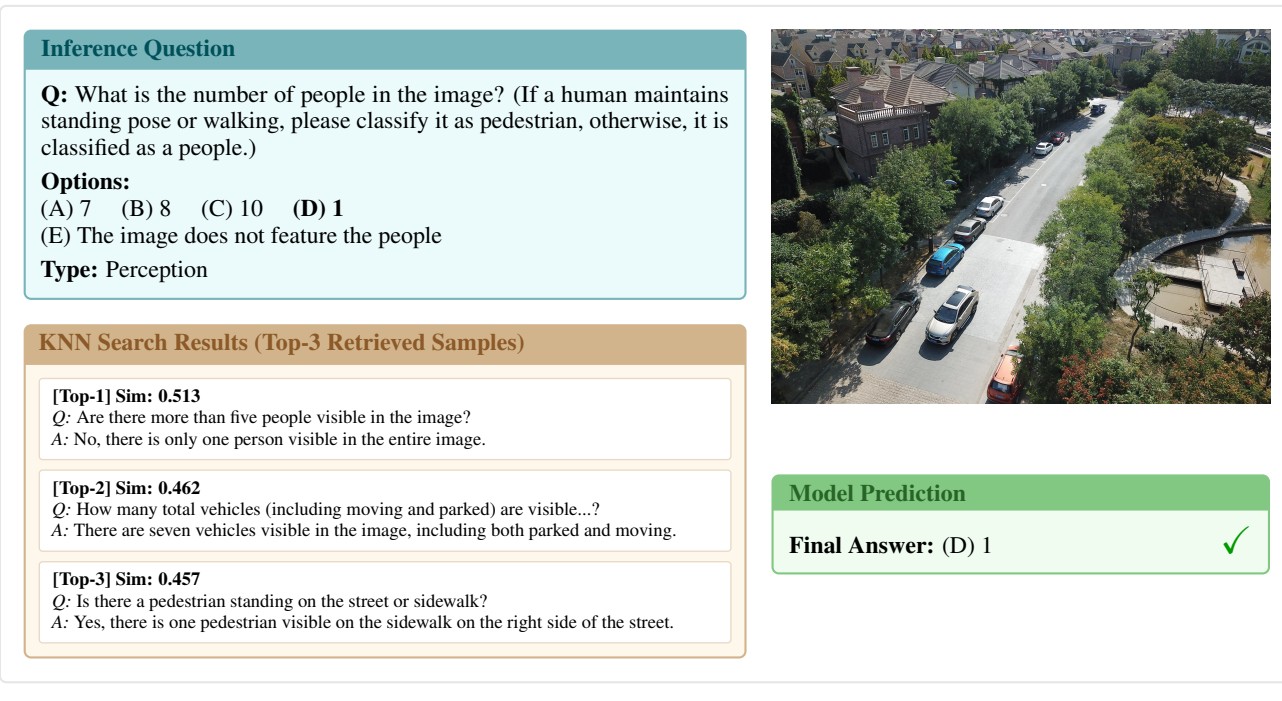

**Inference Question**

**Q:** What is the number of people in the image? (If a human maintains standing pose or walking, please classify it as pedestrian, otherwise, it is classified as a people.)

**Options:**
(A) 7   (B) 8   (C) 10   **(D) 1**
(E) The image does not feature the people

**Type:** Perception

**KNN Search Results (Top-3 Retrieved Samples)**

**[Top-1] Sim: 0.513**
*Q:* Are there more than five people visible in the image?
*A:* No, there is only one person visible in the entire image.

**[Top-2] Sim: 0.462**
*Q:* How many total vehicles (including moving and parked) are visible...?
*A:* There are seven vehicles visible in the image, including both parked and moving.

**[Top-3] Sim: 0.457**
*Q:* Is there a pedestrian standing on the street or sidewalk?
*A:* Yes, there is one pedestrian visible on the sidewalk on the right side of the street.

**Model Prediction**

**Final Answer:** (D) 1 ✓

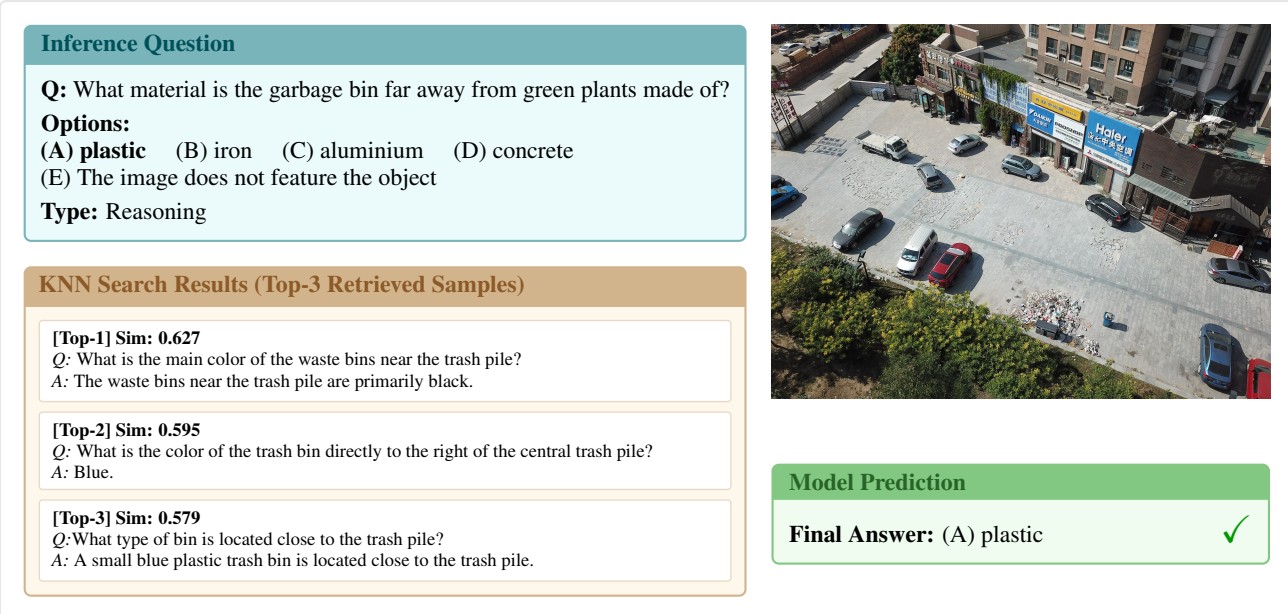

**Inference Question**

**Q:** What material is the garbage bin far away from green plants made of?

**Options:**
**(A) plastic**   (B) iron   (C) aluminium   (D) concrete
(E) The image does not feature the object

**Type:** Reasoning

**KNN Search Results (Top-3 Retrieved Samples)**

**[Top-1] Sim: 0.627**
*Q:* What is the main color of the waste bins near the trash pile?
*A:* The waste bins near the trash pile are primarily black.

**[Top-2] Sim: 0.595**
*Q:* What is the color of the trash bin directly to the right of the central trash pile?
*A:* Blue.

**[Top-3] Sim: 0.579**
*Q:* What type of bin is located close to the trash pile?
*A:* A small blue plastic trash bin is located close to the trash pile.

**Model Prediction**

**Final Answer:** (A) plastic ✓

*Figure S4.* KNN retrieval results for the inference question. We display the top-3 most similar QA pairs retrieved from the training set based on the query embedding.

---

**Prompt for Multi-level Video Captioning**

**System Role & Instructions:**

You are a professional video annotator. Based on the input video clip, generate three levels of natural, fluent, and descriptive captions in English.
**Output Format:** Please output a JSON with fields: `"caption_high_level"`, `"caption_mid_level"`, `"caption_low_level"`.

**Guidelines:**

- Capture temporal dynamics (movements, actions, transitions).
- Describe scene evolution and remain factual.
- Handle coarse captions by translating factual entities to English and ignoring irrelevant expressions.

**Caption Levels:**

- **Level 1 ($<$32 tokens):** A very short summary of the main subject.
- **Level 2 (32-128 tokens):** A richer description covering entities, actions, and context.
- **Level 3 (128-256 tokens):** A comprehensive narrative detailing attributes, spatial relations, and temporal evolution.

**Few-shot Example:**

**User Input (Example):** `[Example Video Frames...]`

**Assistant Output (Example):** `{"caption_high_level": "...", ...}`

---

**Prompt for Triple-Level Image Captioning**

**System Role:**

You are a professional image annotator. Based on the input image (and optionally a coarse caption), generate three levels of natural, fluent, and descriptive captions in English.

**Guidelines:**

**Handling the Coarse Caption**

- The coarse caption may be in **any language**. Always **translate or normalize** it into English concepts.
- Ignore any **emotional, vague, or irrelevant expressions**. Do **not** copy non-English text unless it is **visually present**.

- **Level 1 Caption ($<$32 tokens):** Very short and concise, focusing only on the **main subject or scene**.
- **Level 2 Caption (32-128 tokens):** An **extended but still compact** description. Add **object attributes**, **basic spatial layout**, and **environmental context**.
- **Level 3 Caption (128-256 tokens):** A **fully detailed narrative**. Include: **fine-grained attributes**, **spatial relationships**, **object interactions**, and **environmental details**.

*Figure S5.* Prompt used for multi-level video and image captioning.

**Two-Stage Prompt for Entity Registry Generation**

**Step 1: Entity Extraction per Clip**

**System Role & Instructions:**

You are a professional video entity extractor. Analyze the input video clip and detect all entities. For each entity, provide its name, appearance, identity, and the first seen timestamp within this clip. Avoid duplicates and output a JSON object.

**Few-shot Example:**

**User Input (Example):** `[Video Frames]` with start time `00:00:04`.

**Assistant Output (Example):**

```
{
  "entities": [
    {
      "name": "Dog_A", "appearance": [...],
      "identity": [...],
      "first_seen": "00:00:04"
    },
    ...
  ]
}
```

↓

**Step 2: Global Entity Merging**

**System Role & Instructions:**

You are a professional assistant for merging entities. Your input is a list of entity registries from different clips of the same video. Merge duplicate entities, keep the earliest first seen time, and create a timestamps field to store all occurrence times.

**Few-shot Example:**

**User Input (Example):** A list of entities, e.g.,
`["name": "Dog_X", ..., "name": "Dog_Y", ..., ...]`

**Assistant Output (Example):**

```
{
  "entities": [
    {
      "name": "Dog_A", ...,
      "first_seen": "00:00:00",
      "timestamps": ["00:00:00", "..."]
    },
    {
      "name": "frisbee_A", ...,
      "first_seen": "00:00:04",
      "timestamps": ["00:00:04", "..."]
    },
    ...
  ]
}
```

*Figure S6.* Prompt used for entity generation and merging in a two-stage manner.

---

**Prompt for Clip-level Video QA Generation**

**System Role & Instructions:**

You are a careful dataset constructor that converts one video-clip into high-quality open-ended Q&A pairs. Given a video clip with multi-level captions and entities, produce a diverse set of Q&A that fully covers the clip across scales and aspects.

**Guidelines:**

**Question Composition:**

- **Content Granularity:** High-level (gist), Mid-level (attributes, relations), Low-level (fine-grained details).
- **Temporal Granularity:** Short (single moment), Medium (sequence), Long (multi-event).

**Timestamp Rules:**

- Must be within clip's time range.
- Use float seconds.
- Prefer ranges for sustained actions.
- Must be included in the question text.

**Hints (Workflow):**

1. Parse all captions to collect atomic facts.
2. Resolve facts into a fused, contradiction-free set.
3. Draft questions spanning all granularities.
4. Enforce timestamp rules.
5. Filter for unique, answerable questions.
6. Output JSON only.

**Few-shot Example:**

**User Input (Example):**

```
{
  "caption_high_level": "Chefs prepare food in a busy professional kitchen.",
  "caption_mid_level": "Inside a bustling commercial kitchen, multiple chefs
                        clad in white...",
  "caption_low_level": "In this dynamic video sequence within a professional
                        kitchen, chefs...",
  "clip_begin_time": "205.0", "clip_end_time": "210.0",
  "entities": [{"name": "chef_A", ...}, {"name": "chef_B", ...}]
}
```

**Assistant Output (Example):**

```
{
  "qa": [
    {
      "question": "Between 205.0 – 210.0, what environment is depicted?",
      "answer": "A busy professional kitchen."
    },
    {
      "question": "What are the chefs wearing during 205.0 – 210.0?",
      "answer": "White shirts and black aprons."
    },
    ...
  ]
}
```

*Figure S7.* Prompt for generating clip-level video QA pairs.

**Prompt for Image QA Generation**

**System Role:**

You are a careful dataset constructor that converts an image into a diverse set of 150-200 high-quality, open-ended Q&A pairs with a strong emphasis on low-level perception. Base all Q&A strictly on visual evidence.

**Core Guidelines & Strategies:**

**Perception Priority:**

- **Objects:** Presence, categories, parts.
- **Attributes:** Colors, materials, textures, states.
- **Layout:** Relative positions, depth, orientation.
- **Counting:** Total and conditional counts.
- **OCR:** Transcribe all visible text exactly.
- **Details:** Logos, icons, UI elements, occlusion.

**Perception-Focused Patterns:**

- **Presence:** "Is there a [object]...?"
- **Attribute:** "What color/material is the [object]?"
- **Relation:** "What is to the left of [object]?"
- **Counting:** "How many [objects] are there?"
- **Part-based:** "Does the [device] have a handle?"
- **OCR:** "What text is on the [sign]?"
- **Pose:** "Which way is the [person] facing?"

**Question Diversity:**

- **Types:** What, Where, How many, etc.
- **Reasoning:** Multi-hop, causal, comparative.
- **Variations:** Paraphrase facts into multiple question forms.

**Negative Examples (˜15%):**

- **Format:** Ask a question with a plausible but false premise.
- **Answer:** Start with "No," then provide the correct fact.
- **Errors:** Wrong attributes, counts, relations, etc.

**Systematic Coverage Strategy:**

1. Extract all entities and their attributes.
2. Extract all actions and spatial relations.
3. Extract all visible text.
4. Cross-reference to create complex questions.

**Few-shot Example Snippet:**

**User Input:** An image is provided.

**Assistant Output (Example):**

```
{
  "qa": [
    { "question": "How many people are visible?", "answer": "Three." },
    { "question": "What color is the sweater on the left?", "answer": "Beige." },
    { "question": "Is the building on the right made of brick?",
      "answer": "No, it is made of concrete." },
    ...
  ]
}
```

*Figure S8.* Prompt for generating image QA pairs.

---

**Prompt for Video-level QA Generation (General Instructions)**

**Goal:**

You are a helpful teacher assistant to teach others to exploring, reasoning and understanding a long video. To achieve this goal, your task is to generate a set of questions with answers, based your comprehensive understanding of the video. You must ensure each question is unique and covers different aspects of the video. Each question you generated should be a multiple choice question with four choices, with only one as correct answer and three distractors.

**Core Directives:**

**Strict Rules:**

- Plan extensively before each function call, and reflect on outcomes.
- Only use arguments from user or prior function outputs.
- Inspect video content with tools; do not guess.
- Timestamps can be 'HH:MM:SS' or 'MM:SS'.

**Tools:**

- `global_browse`: Get global information on events/subjects.
- `clip_search`: Search without a specific timestamp.
- `temporal_search`: Search with a timestamp.
- `frame_inspect`: Get fine-grained detail for a time range.

**Suggested Workflow (per question):**

1. Review previous questions for diversity and coverage.
2. Collect information for the target question using tools.
3. Synthesize a question with 4 options (1 correct, 3 plausible distractors).
4. Confirm the question and choices follow design principles.
5. Assign a difficulty level (easy, medium, hard).
6. Call "finish" tool to output the result.

**Output Format:**

Each question must have the following fields:

```
{
  "question": "The question text in plain text format.",
  "options": { "A": "Option A", "B": "Option B", ... },
  "answer": "A single letter of the correct answer label (A, B, C, or D).",
  "clue_duration": [["HH:MM:SS", "HH:MM:SS"], ...],
  "distractor_rationales": { "B": "Rationale for why B is wrong.", ... },
  "difficulty": "'easy', 'medium', or 'hard'"
}
```

*Figure S9.* Prompt for generating video level QA. Note this is only the general instruction prompt. The category-specific instructions are demonstrated in the following figures.

---

**Prompt for Video-level QA Generation (Category-Specific Instructions)**

**Category 1: Temporal Grounding**

Now, please generate high-quality multiple-choice Q&A pairs that test the ability of temporal grounding (locating events on the timeline, ordering, and durations).
**Question Design Principles:**

- Ask in forms such as: "What happened at time T?", "How long did X behavior last?", "What happened immediately after event A?", etc.
- Ensure the coverage of the question sets; do not generate questions that are too similar to each other.
- Do not include the answer text in the question.
- The question must not be answerable by common sense alone.
- The four options must be abstracted at **comparable granularity** and similar length; grounded in nearby or similar scenes, but differ in crucial timing/action details.

**Instructions:**

1. Use `global_browse_tool` to build a global outline of the video's events.
2. Use `clip_search_tool` over the whole video to find candidate moments or intervals that instantiate time points, durations, or order (before/after/next).
3. For each candidate, Use `frame_inspect_tool` (possibly multiple times) to verify the exact time(s) and local context. Identify the minimal evidence interval(s) and record them as `clue_duration`.

**Category 2: Summarization**

Now, please generate high-quality multiple-choice Q&A pairs to test the ability of summarization (themes, key event developments and stage-level evolution).
**Question Design Principles:**

- Ask in forms such as: "Which of the following best reflects the main objective/theme of this video?", "Which of the following best summarizes the core development in section X?", etc.
- Time expressions can be in HH:MM:SS or relative ordering/duration.
- Ensure the coverage of the question sets; do not generate questions that are too similar to each other.
- Do not include the answer text in the question.
- The question must not be answerable by common sense alone.
- The four options must be abstracted at **comparable granularity** and similar length; distractors should reflect "theme drift / missing a crucial phase / overemphasizing minor details."

**Instructions:**

1. Use `global_browse_tool` to map the video's storyline/chapters.
2. Use `clip_search_tool` to extract candidate evidence across segments.
3. Use `frame_inspect_tool` within each involved segment to pin down key frames.

**Category 3: Reasoning**

Now, please generate high-quality multiple-choice Q&A pairs that test the ability of multi-step reasoning (cause effect, intentions, emotions, or plausible prediction grounded in observed evidence).
**Question Design Principles:**

- Ask in forms such as: "Why did the experiment fail?", "What most likely motivated X to do Y?", "Given Z happened, what is the most reasonable immediate follow-up?", etc.
- Time expressions can be in HH:MM:SS or relative ordering/duration.
- Ensure the coverage of the question sets; do not generate questions that are too similar to each other.
- Do not include the answer text in the question.
- The question must not be answerable by common sense alone.
- The four options must be abstracted at **comparable granularity** and similar length; distractors should be **plausible but contradicted** by a missing or conflicting logical chain.

**Instructions:**

1. Use `global_browse_tool` to propose candidate reasoning chains (e.g., cause → intermediate → outcome; intention → action → consequence).
2. Use `clip_search_tool` to locate each chain's key nodes.
3. Use `frame_inspect_tool` to verify each node.

*Figure S10.* Category-specific instructions for generating video-level QA pairs. These prompts are used in conjunction with the general instructions shown in Figure S9.

**Prompt for Video-level QA Generation (Additional Category-Specific Instructions)**

---

**Category 4: Entity Recognition**

Now, please generate high-quality multiple-choice Q&A pairs that test the ability of entity recognition (identifying, distinguishing, and tracking entities, i.e., people/objects/places).
**Question Design Principles:**

- Ask in forms such as: "What color is the clothing of the person arguing with A in the hallway?", "Who performs action Z (by jersey number/apparel feature)?", etc.
- Time expressions can be in HH:MM:SS or relative ordering/duration.
- Ensure the coverage of the question sets; do not generate questions that are too similar to each other.
- Do not include the answer text in the question.
- The question must not be answerable by common sense alone.
- The four options must be abstracted at **comparable granularity** and similar length; distractors should be **visually similar entities** in the same setting but differing in key attributes (color/number/position).

**Instructions:**

1. Use `global_browse_tool` to list principal entities (people/objects/locations).
2. Use `clip_search_tool` to find discriminative cues (apparel color, number, accessories, spatial position).
3. Use `frame_inspect_tool` to confirm decisive visual features for the referenced entity.

---

**Category 5: Event Understanding**

Now, please generate high-quality multiple-choice Q&A pairs that test the ability of event understanding (understanding event-level semantics, i.e., stage/type discrimination, and major scene transitions).
**Question Design Principles:**

- Ask in forms such as: "Which description best characterizes this phase?", "Which stage follows event A?", etc.
- Time expressions can be in HH:MM:SS or relative ordering/duration.
- Ensure the coverage of the question sets; do not generate questions that are too similar to each other.
- Do not include the answer text in the question.
- The question must not be answerable by common sense alone.
- The four options must be abstracted at **comparable granularity** and similar length; distractors should be **nearby types/phases** that look similar but are inconsistent with the confirmed evidence.

**Instructions:**

1. Use `global_browse_tool` to sketch the event timeline and phase segmentation.
2. Use `clip_search_tool` to focus on turning points or scene switches.
3. Use `frame_inspect_tool` to confirm diagnostic visuals (prop/venue change, audience reaction, scoreboard state, etc.).

---

**Category 6: Key Information Retrieval**

Now, please generate high-quality multiple-choice Q&A pairs that test the ability of key information retrieval (**extracting precise visual details** such as slides, screens, scoreboards, documents).
**Question Design Principles:**

- Ask in forms such as: "What quarterly revenue growth is shown on the slide?", "What is the scoreboard at time T?", etc.
- Time expressions can be in HH:MM:SS or relative ordering/duration.
- Ensure the coverage of the question sets; do not generate questions that are too similar to each other.
- Do not include the answer text in the question.
- The question must not be answerable by common sense alone.
- The four options must be abstracted at **comparable granularity** and similar length; distractors should be **nearby types/phases** that look similar but are inconsistent with the confirmed evidence.

**Instructions:**

1. Use `global_browse_tool` to select the target entity or events with rich details.
2. Use `clip_search_tool` to find segments containing visual details, such as on-screen text, digits, tables, charts, scoreboards, or reports.
3. Use `frame_inspect_tool` to ensure readability and confirm the exact detail(s).

---

*Figure S11.* Additional category-specific instructions for generating video-level QA pairs. These prompts supplement those in Figure S10.

