# OpenReview forum: "From Content to Knowledge: Lightning Fast Long-Video Understanding with Neural Knowledge Representations"
_ICML.cc/2026/Conference — ICML 2026 regular_

### Official Review · Reviewer_KntB · 2026-02-18

**Soundness:** 1
**Presentation:** 2
**Significance:** 2
**Originality:** 1
**Overall Recommendation:** 2
**Confidence:** 4

**Summary:**

This work explores an efficient video representation method NKR, which represent video contents as an individual small portion of network weights attached to the VLM backbone. This method relies external VLMs to generate dense captions and question-answering pairs as the knowledge extraction for the video. Experiments on two long-video benchmarks show the effectiveness.

**Compliance With Llm Reviewing Policy:**

Affirmed.

**Final Justification:**

I appreciate the authors' efforts, but I maintain my rating for two reasons.

1. Unfair time comparison

To encode a video into model parameters, the proposed method requires calling APIs and agent workflows to construct data for online training. The time overhead of this real-time process should not be ignored during time comparison. In addition, the manuscript stated that "Existing paradigms treat the video ... that must be repeatedly opened", but during discussion, they reclaim that they do not assume that "existing methods re-encode the entire video for every new question."

2. Limited novelty and generalization

This method represents videos as text and then training the model to memorize the text, rather than directly distilling the video content into the model's parameters. I raise this concern because the current pipeline relies on extensive and fragmented prompt designs, including specific prompts for video summarization and various question generation. As shown in the Appendix, the prompts are highly tailored for the specific six categories in LVBench, and the authors rejected to provide results on other video benchmarks during rebuttal. Thus, I believe it is more akin to prompt engineering rather than a new general paradigm.

**Key Questions For Authors:**

This paper presents an insightful motivation. However, the proposed solution is naive and lacks sufficient justification, making the narrative somewhat overclaimed. The specific concerns are listed as follows:

1. Using only textual information to comprehensively represent long videos is impractical. While compressing video content into model parameters is a promising exploration, the method proposed in this work only encodes textual descriptions rather than the video content itself. The information gap bewteen a video and the caption is non-negligible. Consequently, the method may only be effective when the generated descriptions or QA-pairs already cover the user queries, which is unrealistic for real-world scenarios. Notably, the QA-pairs are generated by DeepVideoDiscovery-o3, which is exactly the SOTA on the target benchmark. Although the author partially acknowledges this textual limitation in the paper, it remains a significant drawback. More evaluations on diverse fine-grained video understanding benchmarks such as Long-RVOS[1] and V-Star[2] can help demonstrate the robustness.

2. The time comparison is unfair. For a user-provided video, this method requires training the model for several epochs, along with additional time for data preparation. This latency cannot be neglected in practical deployment. Moreover, for current VLMs, it is  unnecessary to repetitively encode the video for new queries, due to the existence of KV-cache.

3. For fairness, the baseline comparisons should include Qwen3-VL. Furthermore, a baseline that directly integrates the descriptions and QA-pairs as extra prompts should also be included to justify the necessity of additional training.

4. Regarding technical novelty, the proposed method can be essentially summarized as using text to describe video content and subsequently using that text to train the model. It differs significantly from implicit representation methods like NeRF.

[1] Liang, Tianming, et al. "Long-rvos: A comprehensive benchmark for long-term referring video object segmentation." arXiv preprint arXiv:2505.12702.
[2] Cheng, Zixu, et al. "V-star: Benchmarking video-llms on video spatio-temporal reasoning." arXiv preprint arXiv:2503.11495.

**Limitations:**

yes

**Strengths And Weaknesses:**

1. This paper is well-motivated, and truly points out the limitations of current video understanding paradigms.
2. The paper is well-written.

---

> ### Author Rebuttal · Authors · 2026-03-31
>
> We sincerely thank you for the in-depth feedback! We highly value each of your comments, and all your concerns are addressed point by point:
>
> ---
> **Q1. Clarification on Textual Representation and Generalization**
> - (1) Our goal is to show that long-video knowledge can be distilled into a compact neural representation for interactive QA, rather than to solve all fine-grained video tasks. Benchmarks that require pixel-level spatio-temporal mask prediction are not aligned with our current settings. Evaluation on these tasks would require substantial architectural changes or extra supervision on the frozen VLM backbone which is beyond the scope of this paper.
> - (2) NKR does not only works when the synthesized data explicitly covers the test questions:
>   - (A) (**Table R4.1** below) directly feeding the same synthesized descriptions and QA pairs as in-context text performs substantially worse than NKR; and
>   - (B) (**Fig S4. in appendix**) nearest-neighbor analysis shows the benchmark questions are related to, but not identical to, the synthesized QA pairs.
>   - These results suggest that NKR learns a more generalizable internal representation rather than merely recalling matched text.
>
>   *Table R4.1: Performance comparison between the in-context baseline and NKR paradigm. Note that the Qwen2.5 models are trained using 350 text samples, whereas the Qwen3 models utilize only 150 text samples.*
>   | Method | Perception| Reasoning | Overall |
>   | :---: | :---: | :---: | :---: |
>   | Qwen2.5-In_Context-7B | 22.1 | 17.2 | 19.6 |
>   | Qwen2.5-NKR-7B-r32 | 38.4 | 32.5 | 35.4 |
>   | Qwen3-In_Context-8B | 24.4 | 9.3 | 17.6 |
>   | Qwen3-NKR-8B-r32 | 31.9 | 46.4 | 38.4 |
> ---
> **Q2. Question about time comparison.**
> - Our method is not intended for cold-start or one-off scenarios. Instead, NKR targets persistent long videos that are queried repeatedly over time, such as lecture archives, sports replays, or long-form interviews, where the one-time offline cost can be amortized over many future interactions. For a 1-hour video, the offline cost of NKR is amortized after approximately 300 queries compared to Qwen3-VL-8B, and after roughly 90 queries compared to DVD. Under a modest assumption of 4–5 follow-up questions per session, this corresponds to roughly 60–75 sessions and 18–23 sessions, respectively. We believe these are meaningful thresholds for our target scenarios. More importantly, even before full amortization, NKR operates in a different latency regime: after the one-time offline step, subsequent queries can be answered near-instantly, whereas agentic systems may still require tens of seconds or minutes per interaction.
> ---
> **Q3. Comparison with Qwen3-VL**
> - Our results suggest that NKR is capacity-dependent rather than uniformly beneficial at every scale. As reported in the main text, Qwen3-NKR-8B achieves 39.5% on LVBench, lower than the corresponding direct-video baseline Qwen3-VL-8B (53.5%), indicating that under smaller backbones, direct token-based video understanding can remain stronger. Scaling the backbone from Qwen3-8B to Qwen3-14B improves NKR substantially (39.5% → 48.8%), suggesting that backbone capacity is the key factor for effectively internalizing distilled long-video knowledge. Moreover, although not a matched-family comparison, Qwen3-NKR-14B-r256 is already competitive with, or slightly better than, the much larger direct-video baseline Qwen2.5-VL-32B on long-video benchmarks (48.8 vs. 47.6 on LVBench; 52.9 vs. 48.4 on LongVideoBench-Long), while operating at much lower inference cost. We therefore view NKR not as a universal drop-in replacement at every scale, but as a capacity-dependent efficiency trade-off that becomes attractive with a sufficiently strong backbone.
> ---
> **Q4. Regarding technical novelty**
> - We agree that NKR is not an implicit representation in the classical NeRF sense. Our contribution is instead to adopt the implicit-representation viewpoint for long-video understanding: can a video’s knowledge be materialized into a compact set of parameters, rather than being repeatedly accessed as tokens or through an external database? In this sense, the novelty of our work lies in proposing a new representation-and-inference paradigm for long-video understanding, where a video is compiled into a small, swappable adapter that can be mounted onto a frozen backbone and queried directly at runtime. We also argue that the summary of (“text then train”) understates our contribution. Table R4.1 shows that directly using the same synthesized text as in-context input performs much worse than NKR, and nearest-neighbor analysis (Fig S4. in the appendix) suggests the test questions are related to, but not duplicates of, the synthesized QA pairs. Moreover, NKR remains much more stable than prior methods as video length increases. These results suggest that the benefit comes not from simply storing text, but from learning a more effective internal representation for long-video understanding.

---

> > ### Author Rebuttal · Reviewer_KntB · 2026-04-03
> >
> > The rebuttal fails to address my main concerns. First, the scope of this work is narrow. While the work aims to address long-video challenges, they only consider simple scenarios where the users merely ask high-level questions. Second, even for this target QA scenario where queries repeat multiple times for one video, this work fails to demonstrate advantages over existing methods. As mentioned in my initial comments, due to the existence of caching techniques like prefilling or KV-cache, it is not required to repeatedly encode the video. Third, I don't consider this to be a new technique for video understanding. The model is still updated by text, rather than the video itself. Moreover, this is an existing paradigm for test-time training, while the authors claim their motivation comes from NeRF.

---

> > > ### Author Response · Authors · 2026-04-07
> > >
> > > Thanks for the response. We believe there are still several misunderstandings, and we clarify them below.
> > >
> > > (1) We respectfully believe that the characterization of our scope as a “narrow scenario of high-level questions” is inaccurate. Our paper does not introduce an ad hoc “simple scenario” in which users merely ask high-level questions; rather, we evaluate on a well-known and well-defined long-video QA benchmark used by the community. Moreover, LVBench is not limited to coarse, summarization-style questions. It also contains questions that require recognition of visual details and multi-step reasoning. For this reason, we do not think it is accurate to describe the target setting as one where users “merely ask high-level questions.”
> > >
> > > (2) We believe the reviewer may still misunderstand both the latency comparison used in the paper and our target scenario. Our latency analysis does not assume that existing VLM-based methods re-encode the entire video for every new question. Instead, the reported timing is already considered in an amortized, average-per-query manner, which is precisely why the appeal to prefilling or KV-cache does not address our actual comparison setup. The response appears to argue against a repeated full-encoding baseline that we did not use. More importantly, our rebuttal explicitly clarified that the target scenario is repeated querying over persistent long videos with strong responsiveness requirements. In this setting, the question is not merely whether repeated full encoding can be avoided, but whether the remaining runtime latency is low enough for interactive use. Even under amortized accounting, token-based VLM pipelines still operate in a substantially slower query-time latency regime than NKR.
> > >
> > > (3) Whether a method is novel is not determined solely by whether the immediate supervision signal is text or raw video, but by the representation it learns, the problem formulation it introduces, and the inference regime it enables. In our case, the contribution lies in proposing a new paradigm for long-video QA, where a video’s knowledge is compiled into a compact, swappable parameter adapter that can be queried directly at runtime. More importantly, if the method were merely “updating the model with text,” then directly providing the same synthesized descriptions and QA pairs as in-context input should be competitive. Empirically, it is substantially worse than NKR. This is precisely why we argue that the benefit comes not from text exposure alone, but from internalizing long-video knowledge into a compact parameterized representation.
> > >
> > > (4) Our paper never claims that NKR is a NeRF-style implicit representation of raw video signals, nor that it should be methodologically identical to classical implicit representation methods. Rather, the reference to NeRF is intended to motivate an implicit-representation viewpoint for long-video understanding: whether a video’s knowledge can be materialized into a compact parameterized form, instead of being repeatedly accessed as tokens or through an external database. For our work, the key novelty question is whether prior research has already addressed the same setting for long-video understanding: compiling a persistent long video into a compact, swappable parameter adapter for repeated future QA with near-instant runtime interaction. To the best of our knowledge, this specific setting has not been studied before.

---

### Official Review · Reviewer_qwf7 · 2026-03-11

**Soundness:** 3
**Presentation:** 3
**Significance:** 3
**Originality:** 3
**Overall Recommendation:** 4
**Confidence:** 4

**Summary:**

This paper proposes a new paradigm for long-video understanding by treating a long video as a form of Neural Knowledge Representation. Unlike traditional approaches that feed videos as token streams or construct external retrieval databases, this method compresses and consolidates the semantic content of a video into a small set of additional network weights attached to a VLM backbone.

To achieve this, the authors design an offline Agentic Knowledge Distillation process, leveraging large models to automatically generate dense video captions and high-quality QA pairs to train and fit these weights. During inference, the system simply loads the lightweight NKR weights onto a frozen VLM to answer user queries, without reloading or reprocessing the original video, thereby achieving extremely low inference latency and zero additional memory overhead.

Overall, the idea is relatively novel, but there are some crucial issues that I would like to further discuss with the authors before determining my final score.

**Compliance With Llm Reviewing Policy:**

Affirmed.

**Final Justification:**

I appreciate the authors for solving my concerns, I will retain my positive score.

**Key Questions For Authors:**

1. In what specific real-world application scenarios do the authors believe that such substantial offline costs can be offset by sufficiently high-frequency online queries, making this approach more cost-effective than directly running a single Agentic RAG pipeline per query?
2. Since the original video frames are no longer accessible at inference time, if a user asks about a subtle visual detail that was overlooked by GPT-4 during the offline AKD data generation stage, will the NKR model honestly respond with “I don’t know,” or will it generate a potentially severe hallucination based on the inherent biases of large models?
3. If a very long movie is extended with an extra 10-minute bonus segment, does NKR support incremental updating, or must the entire video undergo the full multi-hour synthesis and distillation process again?

**Limitations:**

Please see weaknesses.

**Strengths And Weaknesses:**

**Strengths**:
1. NKR has competitive inference speed and minimal overhead. Once NKR training is completed, inference requires neither processing any video tokens nor performing database retrieval. This feature can enable stable long-form video understanding.
2. Current methods often exhibit significant performance degradation when handling videos longer than one hour. In contrast, the representation size of NKR is independent of video length, and its performance remains stable across both short and long videos.
3. Experiments demonstrate that NKR is more conducive to holistic reasoning rather than fragmented retrieval, and this characteristic is valuable.

**Weaknesses**:
1. The paper shifts nearly all computational complexity to the offline stage. Processing a one-hour video requires first spending 2–3 hours using GPT-4–level models to synthesize training data, followed by another 2 hours of training on four A100 40G GPUs. This means that for any new video, the system requires at least 4–5 hours of preparation before it can be queried “instantly” by users. Such a setup is impractical for real-time streaming or fast-paced news scenarios.
2. The AKD process relies entirely on converting videos into pure textual descriptions for distillation. This cross-modal lossy compression may result in the loss of fine-grained visual information that is difficult to precisely express in language, such as complex spatial trajectories, subtle motion details, and color gradients.
3. Once trained, NKR becomes fixed. If the agent overlooks an inconspicuous background object during the distillation stage, the resulting NKR will completely lack knowledge about that object. Since the system discards the original video at inference time, it cannot “revisit” the source material for secondary verification as RAG-based methods do. As a result, it may be more prone to irrecoverable hallucinations when faced with adversarial or unexpected queries.

---

> ### Author Rebuttal · Authors · 2026-03-31
>
> We sincerely thank you for the in-depth feedback! We highly value each of your comments, and all your concerns are addressed point by point:
>
> ---
>
> **Q1. Question about offline computation overhead.**
> - For a 1-hour video, the offline cost of NKR is amortized after approximately 300 queries compared to Qwen3-VL-8B, and after roughly 90 queries compared to DVD [1]. Under a modest assumption of 4–5 follow-up questions per session, this corresponds to roughly 60–75 sessions and 18–23 sessions, respectively. We believe these are meaningful thresholds for our target scenarios. More importantly, even before full amortization, NKR operates in a different latency regime: after the one-time offline step, subsequent queries can be answered near-instantly, whereas agentic systems may still require tens of seconds or minutes per interaction.
>
> ---
>
> **Q2. Limitation of textual representation.**
> - We have explicitly mentioned (as a limitation) that the current NKR pipeline is limited by its reliance on textual supervision, and that fine-grained visual cues that are difficult to verbalize may not be fully preserved in the current implementation. In our paper, we adopt a text-based strategy mainly because, in our present training setup, the current VLMs suitable for NKR training only provide stable text-based outputs as practical supervision signals. That said, conceptually, NKR only requires a mechanism to distill a video’s knowledge into a compact set of parameters attached to the backbone model; it does not inherently assume that such supervision must be textual. In fact, if future unified multimodal foundation models can expose visual-token output available, our AKD pipeline could in principle be extended to incorporate such signals directly.
>
> ---
>
> **Q3. Concern about hallucination answer**
> - To investigate the impact of potential hallucinations generated during the AKD process, we conduct comparative experiments using teacher models with varying capabilities. Specifically, we employ GPT-4o mini, GPT-4.1, and GPT-5.4 to each generate 150 text samples based on the image dataset used in our paper. This setup simulates scenarios where NKR receives distilled knowledge of varying quality. The results, presented in **Table R3.1**, demonstrate that our method maintains strong robustness against variations in the quality of knowledge produced by the teacher model during AKD.
>
>   *Table R3.1: Ablation of NKR performance with different teacher models on the image dataset. All result is reported using QWen3-VL-8B-NKR-r32.*
>   | Teacher Model | Perception| Reasoning | Overall |
>   | :---: | :---: | :---: | :---: |
>   | GPT-4o mini | 37.0 | 44.3 | 40.3 |
>   | GPT-4.1 | 31.9 | 46.4 | 38.4 |
>   | GPT-5.4 | 34.9 | 45.9 | 40.5 |
>
> ---
>
> **Q4. Incremental updating support.**
> - We are currently exploring methods to efficiently support incremental updates. Our early approach including a direct weight merging approach to achieve this without requiring the entire video to undergo the full synthesis and distillation process again.
> Assuming the original video's LoRA is $LoRA_1 = \Delta W_1 = A_1 B_1$ and the new segment's LoRA is $LoRA_2 = \Delta W_2 = A_2 B_2$, we can concatenate them as:
>
>     $$\Delta W_{full} = [A_1, A_2] \begin{bmatrix} B_1 \\ B_2 \end{bmatrix} = \Delta W_1 + \Delta W_2$$
>
>     a truncation with SVD is then performed to preserve rank:
>
>     $$SVD(\Delta W_{full}) = USV^T$$
>
>     $$\Delta W_{merge} = U[:, :r_{new}]S[:r_{new}]V^T[:r_{new}, :]$$
>
> ---
> [1] Deep Video Discovery: Agentic Search with Tool Use for Long-form Video Understanding. NIPS 2025.

---

> > ### Author Rebuttal · Reviewer_qwf7 · 2026-04-01
> >
> > I appreciate the authors for solving my concerns, I will retain my positive score.

---

> > > ### Author Response · Authors · 2026-04-07
> > >
> > > Dear Reviewer qwf7,
> > >
> > > Thank you very much for the positive score! We greatly appreciate the time and effort you have taken to help us improve our work. We will continue to refine and enhance our work accordingly.
> > >
> > > Best regards,
> > >
> > > Authors

---

### Official Review · Reviewer_fgC7 · 2026-03-12

**Soundness:** 2
**Presentation:** 3
**Significance:** 3
**Originality:** 4
**Overall Recommendation:** 4
**Confidence:** 3

**Summary:**

This paper proposes a new paradigm for long-video understanding by representing a single video as a `Neural Knowledge Representation (NKR)` rather than as a long token sequence or an external retrieval database. Concretely, each video is distilled into a LoRA-style adapter attached to a frozen VLM, using an `Agentic Knowledge Distillation (AKD)` pipeline that automatically generates descriptions and QA pairs for offline training. At inference time, the model answers text queries by loading the corresponding NKR, without re-encoding the original video. The reported results suggest that this design offers attractive online efficiency while maintaining competitive accuracy on the evaluated benchmarks.

**Compliance With Llm Reviewing Policy:**

Affirmed.

**Final Justification:**

All concerns are sufficiently addressed with clear, quantitative experiments.
The work is solid and practically meaningful, although its novelty remains moderate.

**Key Questions For Authors:**

1. Could you provide a break-even analysis showing how many queries per video are needed for NKR to amortize its offline cost?
2. Why are there no more tightly controlled baselines with matched backbone and teacher budget?
3. To what extent does the benchmark-aware QA synthesis process affect the reported generalization claims?

**Limitations:**

No. I would encourage the authors to discuss more explicitly the dependence on proprietary teachers, the full offline cost, the risk of benchmark-aware data synthesis, and the implications of storing video knowledge in parameter form for updating or deletion.

**Strengths And Weaknesses:**

Strengths:
- The central idea is original and conceptually interesting.
- The method offers a compelling efficiency perspective for repeated multi-turn querying.
- The overall pipeline is reasonably well aligned, from representation design to distillation and use at inference time.
- The ablations suggest that the different distillation components contribute meaningfully.

Weaknesses:
- A substantial amount of cost is shifted to offline data generation and per-video adapter training.
- The latency claims are less persuasive without incorporating the full offline cost.
- The comparisons would be stronger with more controlled baselines using matched backbones and preprocessing budgets.
- The approach relies heavily on strong proprietary teachers and an agentic distillation pipeline.
- Evidence for broader generalization and real deployment value remains limited.

---

> ### Author Rebuttal · Authors · 2026-03-31
>
> We sincerely thank you for the in-depth feedback! We highly value each of your comments, and all your concerns are addressed point by point:
>
> ---
>
> **Q1. Break-even Analysis of offline computation overhead.**
> - We would like to clarify that NKR is primarily designed for repeated-access long-video scenarios, such as online education, webinar or meeting archives, sports replay analysis, and long-form documentary/interview archives. In these settings, the same video is often queried many times by different users over its lifetime, so the one-time offline compilation cost can be amortized over repeated downstream interactions. To make this more concrete, we conduct a break-even analysis against both token-based and agentic approaches:
> - Compared to token-based methods. We benchmark our approach against Qwen3-VL-8B. Our analysis indicates that for a 1-hour video, the offline cost of NKR is amortized after approximately 300 queries.
> - Compared to agentic methods. Benchmarking against DVD [1], we find that for a 1-hour video, the offline cost is amortized after roughly 90 queries.
>
> To make the query count more interpretable, one can translate it into multi-turn access sessions. In many target scenarios (e.g., lecture archives, webinar recordings, or sports replay analysis), a single access often involves multiple follow-up questions rather than a single query. Under a modest assumption of 4–5 queries per session, a break-even point of 300 queries corresponds to roughly 60–75 sessions, while 90 queries corresponds to roughly 18–23 sessions.
> This is already a meaningful and useful threshold for our target scenarios where a single long video may support many rounds of follow-up questions from multiple user sessions. More importantly, even before full amortization, NKR operates in a fundamentally different latency regime: after the one-time offline step, subsequent queries can be answered near-instantly, whereas agentic systems may still require tens of seconds or minutes per interaction.
>
> ---
>
> **Q2. Comparison with matched backbones.**
> - Our main paper reports an overall accuracy of 39.5% on LVBench using QWen3-8B as backbone, which underperforms the corrsponded QWen3-VL-8B model (53.5%, using maximum of 2048 frames as input). On the other hand, our experiment also shown that our method on Qwen3-14B slightly outperforms Qwen2.5-VL-32B, a significant larger VLM model. These experiments suggest a nuanced picture: NKR is capacity-dependent rather than uniformly beneficial at every scale. Under smaller backbones, direct token-based video understanding can remain stronger, which indicates that NKR is not a free lunch and places a stronger burden on the backbone to internalize and reason over distilled knowledge. At the same time, this should not be interpreted as evidence that the paradigm lacks value. Our ablation shows that increasing the backbone from Qwen3-8B to Qwen3-14B improves NKR performance from 39.5% to 48.8%, suggesting that backbone capacity is the key factor for leveraging distilled long-video knowledge. Moreover, although not a matched-family comparison, our **main and appendix results** show that Qwen3-NKR-14B-r256 is already competitive with, or slightly better than, a much larger direct-video baseline Qwen2.5-VL-32B on long-video benchmarks (48.8 vs. 47.6 on LVBench; 52.9 vs. 48.4 on LongVideoBench-Long), while operating at dramatically lower inference cost. We therefore view NKR not as a universal drop-in replacement for direct video input at every scale, but as a capacity-dependent efficiency trade-off that can shift the performance–latency frontier once paired with a sufficiently capable backbone.
>
> ---
>
> **Q3. Robustness of benchmark-aware QA synthesis**
> - To validate the robustness of the proposed NKR against variations in the QA synthesis process, we employ three models with varying capabilities (GPT-4o mini, GPT-4.1, and GPT-5.4) to generate 150 textual samples each. We then train our model on the image dataset mentioned in the main paper using these synthesized texts. As presented in **Table R2.1**, the results demonstrate that our approach maintains strong robustness across different QA synthesis models.
>
>   *Table R2.1: Ablation of NKR performance with different teacher models on the image dataset. All result is reported using QWen3-VL-8B-NKR-r32.*
>   | Teacher Model | Perception| Reasoning | Overall |
>   | :---: | :---: | :---: | :---: |
>   | GPT-4o mini | 37.0 | 44.3 | 40.3 |
>   | GPT-4.1 | 31.9 | 46.4 | 38.4 |
>   | GPT-5.4 | 34.9 | 45.9 | 40.5 |
> ---
> [1] Deep Video Discovery: Agentic Search with Tool Use for Long-form Video Understanding. NIPS 2025.

---

> > ### Author Rebuttal · Reviewer_fgC7 · 2026-04-04
> >
> > All concerns are sufficiently addressed with clear, quantitative experiments.
> > The work is solid and practically meaningful, although its novelty remains moderate.

---

> > > ### Author Response · Authors · 2026-04-07
> > >
> > > Dear Reviewer fgC7,
> > >
> > > Thank you very much for raising the score! We greatly appreciate the time and effort you have taken to help us improve our work. We will continue to refine and enhance our work accordingly.
> > >
> > > Best regards,
> > >
> > > Authors

---

### Official Review · Reviewer_LGcE · 2026-03-15

**Soundness:** 3
**Presentation:** 3
**Significance:** 2
**Originality:** 3
**Overall Recommendation:** 3
**Confidence:** 4

**Summary:**

The work attempts to address a general topic of high-latency in long-video understanding. It proposes Neural Knowledge Representation (NKR), which transforms video content into a small set of LoRA adapters. Through Agentic Knowledge Distillation (AKD), the model "memorizes" the video, enabling near-instant interaction without the need for long-context tokens or external RAG databases.

**Compliance With Llm Reviewing Policy:**

Affirmed.

**Key Questions For Authors:**

Please refer to Weekness

**Limitations:**

Yes

**Strengths And Weaknesses:**

Strengths:
- Achieves an inference latency of 0.33s (100x speedup) with zero additional runtime memory overhead, as video data is baked into weights rather than stored in KV-Cache.
- An important concept assessed by this study is the decoupling of video length from inference cost. The model maintains stable performance even for videos exceeding one hour.

Weeknesses:
- The work attempts to address a general topic of inference efficiency, but it shifts the heavy computational burden from the inference stage to the pre-processing stage. For a single 1-hour video, the proposed AKD process requires 2-3 hours of agentic data generation and an additional 2 hours of LoRA training on four A100 GPUs. This "training-before-viewing" requirement makes the system impractical for real-time or on-demand video understanding scenarios where users expect immediate interaction after uploading a new video.

- The NKR's "knowledge" is distilled entirely through a textual medium (dense descriptions and QA pairs). Since current VLMs cannot output visual tokens as supervision signals, the model's understanding is limited by what can be articulated in text. Fine-grained visual features (e.g., subtle textures, complex spatial relationships, or non-verbal emotional cues) that are difficult for an agent to describe in words may be permanently lost during the NKR conversion process.

- The paradigm of "one LoRA per video" poses a significant storage and management challenge. While the authors argue that 200MB-500MB per video is small, for a platform with millions of videos, maintaining and indexing millions of separate LoRA adapters is non-trivial. The paper lacks a discussion on how the system would scale or how it would handle the "cold start" problem for a massive, dynamic video repository.

- By training on a small, synthesized dataset (AKD data), there is a high risk of the model "memorizing" incorrect information or hallucinations generated by the teacher agent. If the Agentic Knowledge Distillation produces a wrong fact about a video frame, the NKR will likely bake that error into its parameters. The paper does not sufficiently evaluate how robust the NKR is against "synthetic noise" compared to raw-token-based models.

- An important concept assessed by this study is whether parameter-based memory is superior to context-based memory. However, the evaluation could be strengthened by comparing against more "efficient RAG" or "KV-cache compression" baselines (e.g., StreamingLLM or H2O). Comparing only against vanilla long-context models and basic RAG makes the efficiency gains of NKR look impressive, but it may overlook more optimized, non-training-based alternatives.

---

> ### Author Rebuttal · Authors · 2026-03-31
>
> We sincerely thank you for the in-depth feedback! We highly value each of your comments, and all your concerns are addressed point by point:
>
> ---
>
> **Q1. Question about offline computation overhead.**
> - Our method is not intended for cold-start or one-off scenarios where a user uploads a new video and expects immediate interaction. Instead, NKR targets persistent long videos that are queried repeatedly over time, such as lecture archives or sports replays, where the one-time offline cost can be amortized over many future interactions. For a 1-hour video, the offline cost of NKR is amortized after approximately 300 queries compared to Qwen3-VL-8B, and after roughly 90 queries compared to DVD [1]. Under a modest assumption of 4–5 follow-up questions per session, this corresponds to roughly 60–75 sessions and 18–23 sessions, respectively. We believe these are meaningful thresholds for our target scenarios. More importantly, even before full amortization, NKR operates in a different latency regime: after the one-time offline step, subsequent queries can be answered near-instantly, whereas agentic systems may still require tens of seconds or minutes per interaction.
>
> ---
>
> **Q2. Limitation of textual representation.**
> - We acknowledge that the current NKR pipeline relies on textual supervision, so some fine-grained visual cues may not be fully preserved. This mainly reflects the current training interface, since existing VLMs suitable for NKR training provide stable text-based outputs as practical supervision signals. That said, we do not view this as a limitation of the NKR paradigm itself. NKR only requires a mechanism to distill video knowledge into compact parameters; it does not inherently require textual supervision. If future unified multimodal models expose trainable visual-token outputs, our AKD pipeline could naturally extend to such signals.
> ---
>
> **Q3. The challenge of storing and managing NKRs.**
> - For long videos, we are actively exploring more lightweight NKR designs; Meanwhile, as the video length increases, the ratio of the additional storage overhead introduced by NKR becomes progressively lower because the NKR size grows much more slowly than the video itself. For indexing NKRs, we are happy to notice that recent works in related domains have begun to explore RAG-style indexing and retrieval mechanisms for LoRA-like adapters [2]. Inspired by these efforts, we believe that using video descriptions or other metadata as an index to retrieve the corresponding NKR can be a promising direction for future work.
>
> ---
>
> **Q4. Robustness against teacher agents with different performances.**
> - We conduct experiments using Qwen3-VL-8B on the image dataset mentioned in the paper. We respectively used GPT-4o mini, GPT-4.1, and GPT-5.4 as teachers to each generate 150 pieces of textual information, to represent different levels of the amount of incorrect information present in the textual information. The experimental results are shown in **Table R1.1**, which demonstrates that when the text description is dense enough, the NKR possesses strong robustness against hallucinations and incorrect information present in the teacher.
>
>   *Table R1.1: Ablation of NKR performance with different teacher models on the image dataset. All result is reported using QWen3-VL-8B-NKR-r32.*
>   | Teacher Model | Perception| Reasoning | Overall |
>   | :---: | :---: | :---: | :---: |
>   | GPT-4o mini | 37.0 | 44.3 | 40.3 |
>   | GPT-4.1 | 31.9 | 46.4 | 38.4 |
>   | GPT-5.4 | 34.9 | 45.9 | 40.5 |
> ---
>
> **Q5. Comparison with KV-cache compression method**
> - AdaReTaKe [3] in **Table 2 of the paper** is indeed a KV-cache compression method. However, it still processes video content whose effective information grows with length, and thus remains subject to scalability trade-offs between compression, information loss, and computation.
> By contrast, NKR avoids processing a growing video representation at inference time. Consistent with this, **Table 3 of the paper** shows that NKR is more stable than KV-cache compression methods as video length increases.
>
> ---
> [1] Deep Video Discovery: Agentic Search with Tool Use for Long-form Video Understanding. NIPS 2025. \
> [2] Parametric Retrieval Augmented Generation. SIGIR 2025. \
> [3] AdaReTaKe: Adaptive redundancy reduction to perceive longer for video-language understanding.

---

### Decision · Program_Chairs · 2026-04-30

**Decision:**

Accept (regular)

**Comment:**

The paper proposes Neural Knowledge Representation (NKR) through Agentic Knowledge Distillation to encode long-video context directly into LoRA weights. While the inference-time latency reduction is significant, initial concerns were raised by Reviewer LGcE regarding the offline pre-processing bottleneck. However, the author rebuttal defended this as an amortized cost for heavily re-watched videos. Since there is no clear negative concerns remaining after the rebuttal, the AC recommends weak accept.